# Domain Re-Modulation for Few-Shot Generative Domain Adaptation

**Yi Wu**[*], **Ziqiang Li**[*][†]
University of Science and Technology of China

**Chaoyue Wang**[‡], **Heliang Zheng**, **Shanshan Zhao**
JD Explore Academy

**Bin Li**[‡]
University of Science and Technology of China

**Dacheng Tao**
JD Explore Academy

## Abstract

In this study, we delve into the task of few-shot Generative Domain Adaptation (GDA), which involves transferring a pre-trained generator from one domain to a new domain using only a few reference images. Inspired by the way human brains acquire knowledge in new domains, we present an innovative generator structure called **Domain Re-Modulation (DoRM)**. DoRM not only meets the criteria of *high quality*, *large synthesis diversity*, and *cross-domain consistency*, which were achieved by previous research in GDA, but also incorporates *memory* and *domain association*, akin to how human brains operate. Specifically, DoRM freezes the source generator and introduces new mapping and affine modules (M&A modules) to capture the attributes of the target domain during GDA. This process resembles the formation of new synapses in human brains. Consequently, a linearly combinable domain shift occurs in the style space. By incorporating multiple new M&A modules, the generator gains the capability to perform high-fidelity multi-domain and hybrid-domain generation. Moreover, to maintain cross-domain consistency more effectively, we introduce a similarity-based structure loss. This loss aligns the auto-correlation map of the target image with its corresponding auto-correlation map of the source image during training. Through extensive experiments, we demonstrate the superior performance of our DoRM and similarity-based structure loss in few-shot GDA, both quantitatively and qualitatively. Code will be available at https://github.com/wuyi2020/DoRM.

## 1 Introduction

Domain adaptation aims to bridge the domain gap and transfer knowledge to mitigate the limitations imposed by a lack of extensive labeled data [8; 40; 51]. Recent research has explored the field of few-shot Generative Domain Adaptation (GDA), which aims to achieve realistic and diverse generation with only a few training images [29; 32; 34; 43; 47; 59; 31; 42; 23; 62]. Particularly, the

---

[*]Equal Contribution.

[†]This work was performed when Ziqiang Li was visiting JD Explore Academy as a research intern.

[‡]Corresponding Author.

37th Conference on Neural Information Processing Systems (NeurIPS 2023).

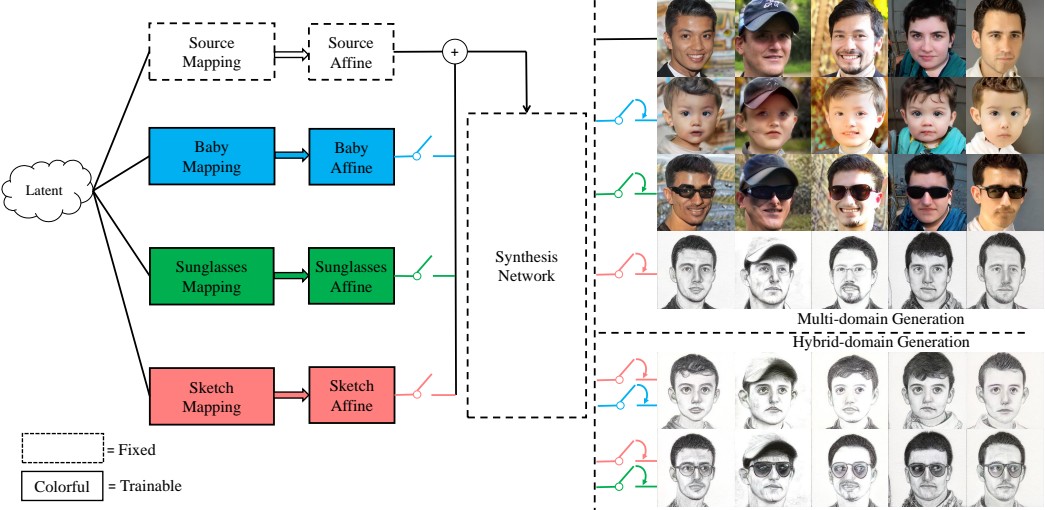

Figure 1: **Left**: Framework of our DoRM. Given a generator of StyleGAN2 structure pre-trained on a source domain (components in the white dotted blocks), we can realize multi-domain generation and hybrid-domain generation by activating the corresponding trained M&A modules (components in the colourful solid blocks). **Right**: Multi-domain generation (right top) and hybrid-domain generation (right bottom). For instance, by activating the trained M&A modules of Baby and Sketch domains, hybrid-domain (Sketch-Baby domain) generation has been realized easily.

purpose of few-shot GDA[4] is to transfer a generator pre-trained on the source domain to the target domain using a few reference images.

Existing few-shot GDA works primarily focus on three properties: *(i) High quality* and *(ii) Large diversity*, where the adapted generator can synthesize high quality and diverse images in the target domain and *(iii) Cross-domain consistency*, where the adapted images and their corresponding source images should be consistent in terms of domain-sharing attributes, such as pose and identity. A majority of them update the entire generator with regularization techniques including GAN inversion [55; 47; 63], Contrastive-Language-Image-Pretraining (CLIP) [10; 55; 63], and consistency loss [32; 47; 59] to achieve large diversity and cross-domain consistency of adaptation. However, updating the entire generator imposes limitations on its ability to synthesize multiple target domains. In contrast, the human brain employs a more efficient approach to learn from new domains by forming new proteins[4], allowing for the retention and integration of knowledge from multiple domains. This creative capacity enables humans to explore previously unseen domains.

Drawing inspiration from human learning processes, we propose a novel generator structure called **Domain Re-Modulation (DoRM)** based on StyleGAN2 [19]. DoRM not only fulfills three essential properties of few-shot GDA but also introduces two new capabilities: *(iv) Memory*, enabling the generator to retain knowledge from previously learned domains when generating images in new domains, and *(v) Domain Association*, allowing the generator to integrate multiple learned domains and synthesize hybrid domains not encountered during training. As depicted in Figure 1, DoRM freezes the pre-trained source generator and introduces new mapping and affine modules (M&A modules) to achieve the necessary domain shift in the style space. This approach enables DoRM to perform high-fidelity generation across multiple domains by selectively activating different M&A modules, which significantly reduces the required storage space. Moreover, as StyleGAN's style space is a linear subspace [36; 24; 45], the domain shift obtained through DoRM is linearly combinable. Consequently, DoRM can synthesize images in hybrid domains not present in the training dataset by simultaneously activating multiple M&A modules. To further enhance cross-domain consistency, we introduce a similarity-based structure loss denoted as $L_{ss}$. Specifically, we leverage the CLIP image

---

[4]It is important to note that the objective of few-shot Generative Domain Adaptation (GDA) differs from that of Few-Shot Image Generation (FSIG). FSIG focuses on generating high-quality and diverse images in the target domain using a limited number of training samples, whereas few-shot GDA involves additional considerations beyond FSIG.

encoder [33] to extract intermediate tokens from the target image and its corresponding source image. We then enforce consistency between the auto-correlation maps of these tokens, ensuring improved alignment between target and source images.

We conducted extensive experiments on few-shot GDA involving various source and target domains. Our experiments, both quantitative and qualitative, highlight the competitive performance of our method compared to state-of-the-art approaches in 10-shot GDA. We achieved remarkable results in terms of quality, diversity, and cross-domain consistency. Notably, our proposed DoRM stands out by demonstrating exceptional performance in hybrid-domain generation, a hard task that previous methods have not accomplished well. The key contributions of this work can be summarized as follows:

- We introduce DoRM, a novel generator structure for few-shot generative domain adaptation, inspired by the learning mechanism of human brains. DoRM not only excels in synthesizing high-quality, diverse, and cross-domain consistent images, but also integrating memory and domain association capabilities that remain relatively unexplored in the field. Notably, our approach is one of the very few that encompasses all five desired properties of GDA.

- Additionally, we propose a novel consistency loss called similarity-based structure loss, which further enhances cross-domain consistency in our approach.

- Through comprehensive evaluations, our proposed method outperforms existing competitors across various settings, showcasing its superior performance.

## 2  Related Work

### 2.1  Generative Adversarial Networks

Deep learning has made remarkable achievements in various fields [46; 44; 22]. Among them, generative adversarial networks [11; 37; 28] play a two-player adversarial game, where the generator aims to synthesize realistic images to fool the discriminator, while the discriminator learns how to distinguish the synthesized images from real ones. Previous works [16; 2; 53; 18; 19] have significantly improved the synthesis capability on high-resolution datasets. Style-based methods [18; 19; 35] remain the state-of-the-art unconditional GANs owing to their unique architecture. However, GANs training requires a large number of training images, or severe discriminator overfitting can occur [26; 27]. To alleviate this issue and improve training stability, various data augmentation techniques [57; 15; 17; 61; 60; 25] and regularization technologies [26; 48; 39; 14] have been introduced. Among these, Adaptive Discriminator Augmentation (ADA) is an adaptive strategy that controls the strength of augmentations and has shown remarkable performance, becoming the default operation in data-efficient GANs. Tseng *et al.* [39] proposed a regularization scheme to modulate the discriminator's prediction, mitigating discriminator overfitting. However, most of these techniques fail in the extremely small (e.g., 10) training data regime.

### 2.2  Few-shot Generative Domain Adaptation

Previous works in few-shot GDA mainly have primarily focused on three essential properties: *High quality*, *Large diversity*, and *Cross-domain consistency*. Some studies have fine-tuned the entire generator using regularization techniques. For example, [32; 47] introduced a consistency loss based on Kullback-Leibler (KL) divergence to preserve the relative similarities and differences between instances, inheriting diversity and maintaining consistency during adaptation. The contrastive term [12; 5] has also been employed to construct consistency loss in [59]. In addition, leveraging the semantic power of large Contrastive-Language-Image-Pretraining (CLIP) [33] models, [10; 55; 63] proposed to define the domain-gap direction in CLIP embedding space, guiding the optimization of the generator. Furthermore, GAN inversion [7; 38] technique has been widely used in few-shot GDA to meet different purposes, such as exploring the domain-sharing attributes [55; 63], and compressing the latent space to a subspace to relax the cross-domain alignment [47]. Compared to optimizing the entire generator, some studies [31] only update the crucial part of the generator. Moreover, some work [49] adds a lightweight attribute adaptor and attribute classifier before the frozen generator and after the frozen discriminator, respectively. The proposed method achieves remarkable performance in synthesis quality and diversity but lacks cross-domain consistency.

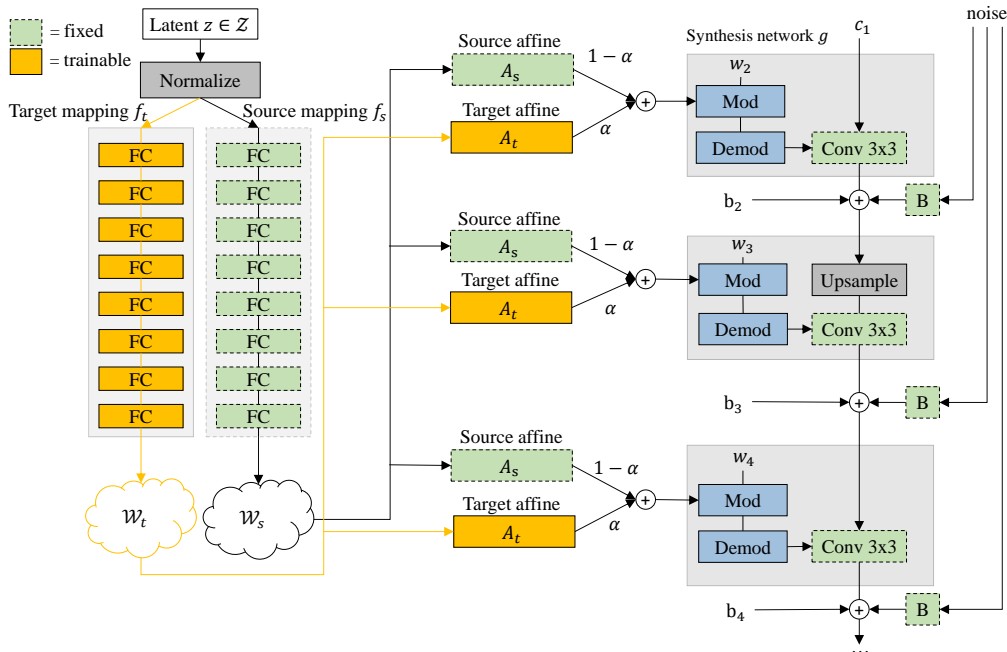

Figure 2: Our framework (**DoRM**) is based on StyleGAN2. In the Synthesis network $g(\cdot)$ (gray blocks), the source affines $A_s$ and the source mapping $f_s$ constitute the pre-trained source generator on a source domain. To achieve generative domain Adaptation, we incorporate a new target mapping $f_t$ and target affines $A_t$. During training, we only optimize the parameters of the solid yellow blocks.

Furthermore, [20; 1] share the similar idea with us of incorporating multi-domain generation capabilities into a single generator. However, these methods employ multiplicative modulation, which is indeed different with ours. Specifically, HyperDomainNet [1] adopts a solitary modulation parameter, primarily a scale ($\delta$), to adjust the weights of the convolutional layer, influencing the "s" space of StyleGAN. In this setup, the scale parameter remains constant for all images within a target domain. Formally, HyperDomainNet's architecture is represented as $w \cdot s_i \cdot \delta$, where $w$ and $s_i$ denote the convolutional weight and style code of the source StyleGAN2, respectively. However, the introduction of this all-encompassing scale parameter can potentially restrict the generator's learning capacity. Especially in scenarios with significant domain gaps between target and source domains, HyperDomainNet's performance may decline. Furthermore, the non-linear combinability of the scale modulation parameter impedes its ability to achieve robust domain association. Among the approaches closest to our DoRM, DynaGAN [20] introduces two modulation parameters—shift ($\Delta s$) and scale ($\delta$)—to the convolutional weights affecting StyleGAN's "s" space. DynaGAN's architecture can be defined as $w \cdot (s_i + \Delta s) \cdot \delta$. Similar to HyperDomainNet, the non-linear combinability of the scale modulation parameter impedes its ability to achieve robust domain association. In contrast, our DoRM solely embraces a sample-specific domain shift denoted as $\Delta s_i$, formalized as $w \cdot (s_i + \Delta s_i)$. This distinct design substantially enhances the generator's learning potential, enabling adaptation to a diverse spectrum of target domains, even in the presence of considerable domain gaps. Notably, the sample-specific domain shift proves sufficient for both few-shot and one-shot GDA, eliminating the necessity for an additional domain scale parameter. This streamlined approach not only fosters domain association but has been effectively demonstrated. Additionally, although [30] defines and addresses a related task called Domain Expansion and domain composition (referred to as memory and domain association in our paper), it inadvertently curtails the source domain's generative capacity, amplifying the intricacies and temporal requisites of domain adaptation. As depicted in the experimental results, [30] encounters challenges when faced with substantial domain gaps between source and target domains. Evidently, its performance is compromised, as evidenced in instances such as adapting to the 10-shot generation context of the Sketch dataset and the one-shot generation context of the "elsa" dataset.

# 3 Approach

In this section, we present our generator structure, DoRM (Section 3.1), which takes inspiration from the way human brains learn new knowledge in different domains. DoRM excels in generating high-quality, diverse images while maintaining strong cross-domain consistency in few-shot GDA. Additionally, it possesses the unique ability to integrate multiple learned domains, allowing for the synthesis of images in hybrid domains that were not present in the training dataset. To address the issue of discriminator overfitting, we freeze the backbone of the source discriminator and introduce a target domain classifier (Section 3.2). Furthermore, we propose a novel similarity-based structure loss, denoted as $L_{ss}$ (Section 3.3), to enhance the maintenance of cross-domain consistency. Finally, we introduce the overall training loss in Section 3.4.

## 3.1 Domain Re-modulation Structure of Generator

**StyleGAN2.** Our method is applied to a pre-trained StyleGAN2 [19]. Unlike general GANs that directly feed latent code to the generator, StyleGAN2 [19] employs a non-linear mapping network $f(\cdot)$ to transform the latent space $\mathcal{Z}$ into an intermediate latent space $\mathcal{W}$. The latent code ($w \in \mathcal{W}$) is transformed into the style code ($s \in \mathcal{S}$) through a learned affine transformation $A$. Finally, the style code is inserted into the synthesis network $g(\cdot)$ through the modulation component at each convolution layer.

**Domain Re-modulation of Generator.** To equip the generator with memory and inherit the semantic information (*e.g.*, glasses, gender, ages) from the source domain, we propose DoRM, illustrated in Figure 2. We freeze the given StyleGAN2 model pre-trained on the source domain and employ a new mapping network $f_t$ and new affine transformation $A_t$ (M&A module) to build $\mathcal{W}_t$ space and $\mathcal{S}_t$ space, respectively. Through the new M&A module, we obtain domain shift as well as the target style codes. The added $f_t$ and $A_t$ have the same architecture as $f_s$ and $A_s$, respectively, and are initialized by $f_s$ and $A_s$. Accordingly, each layer of the synthesis network is controlled by source and target style codes. Given a latent code $z$ sampled from Normal distribution $\mathcal{Z}$ ($z \in \mathcal{Z}$),

$$w_t = f_t(z) \in \mathcal{W}_t, \quad w_s = f_s(z) \in \mathcal{W}_s, \tag{1}$$

where $\mathcal{W}_t$ and $\mathcal{W}_s$ contain information from the target domain and source domain, respectively. Transformed by their own learned affine layers (Eq. 2), the information in these two domains represents their respective styles and is combined into general styles $s$ (Eq. 3) to control the synthesis network together:

$$s_t = A_t(w_t), \quad s_s = A_s(w_s), \tag{2}$$

$$s = \alpha s_t + (1 - \alpha)s_s, \tag{3}$$

where $\alpha$ is a hyper-parameter that controls the strength of the domain shift. In GDA, it tends to reuse various factors like pose, content, structure, *etc.* from the source domain and learns the most distinguishable characteristics of the target domain. To preserve variation factors as much as possible in the source domain, $\alpha$ is set to a relatively small value (More analyses can be found in Section A.1). The combined style code $s$ modulates the convolution weights through $w_i' = s \cdot w_i$, where $w_i'$ is the modulated weights. Then, demodulation is employed to eliminate the influence of the style code $s$ from the statistics of the convolution's output feature maps, which has been formalized as $w_i'' = \frac{w_i'}{\|w'\|_2}$, where $\|w'\|_2$ represents the L2-Norm function.

## 3.2 Target Domain Classifier

In GANs training, the discriminator typically distinguishes the training images from the generated images. However, in few-shot GDA, where there are extremely few training images, the discriminator can easily overfit to the training images, leading to severe model collapse in the generator. In this work, we treat the discriminator as a target domain classifier that measures the probability of the images belonging to the target domain. Defining the target domain by only a few training images is ambitious, so we desire the discriminator to extract the most representative feature of training images that portray the target domain. Inspired by [29], we reuse and fix the feature extractor $d(\cdot)$ of the pre-trained source discriminator to inherit its strong feature extraction capability. Additionally, we introduce a target domain classifier $\phi(\cdot)$ on the top of the feature extractor $d(\cdot)$. We use a two-layer multi-layer perceptron (MLP) as the target domain classifier $\phi(\cdot)$ and the target domain classifier

$\phi(\cdot)$, and it is updated from scratch, which outperforms directly fine-tuning the final layer of the original discriminator, especially for large domain-gap GDA. Given an image $x$, the discriminator measures the probability that the image belongs to the target domain using $p = \phi(d(x))$.

### 3.3 Similarity-based Structure Loss

To explicitly model the cross-domain consistency in generative domain adaptation, we propose a novel similarity-based structure loss called $L_{ss}$. Our intuition is that the auto-correlation maps of the source image and its corresponding target image should be consistent during generative domain adaptation. To achieve this, we extract the intermediate tokens $F_A$ and $F_B$ of the source image $I_A$ and its corresponding target image $I_B$ from the k-th layer pf the CLIP image encoder. These tokens are denoted by $F_A = F_A^1, \cdots, F_A^n$ and $F_B = F_B^1, \cdots, F_B^n$, respectively. We define the auto-correlation maps as $M_A = \frac{F_A^T}{|F_A^T|} \times \frac{F_A}{|F_A|}$ and $M_B = \frac{F_B^T}{|F_B^T|} \times \frac{F_B}{|F_B|}$, where $\boldsymbol{M}_A^{i,j} = \frac{\boldsymbol{F}_A^i \cdot \boldsymbol{F}_A^j}{|\boldsymbol{F}_A^i||\boldsymbol{F}_A^j|}$ and $\boldsymbol{M}_B^{i,j} = \frac{\boldsymbol{F}_B^i \cdot \boldsymbol{F}_B^j}{|\boldsymbol{F}_B^i||\boldsymbol{F}_B^j|}$. The $L_{ss}$ is then defined as the L2 norm of the difference between $M_A$ and $M_B$:

$$L_{ss} = \frac{1}{n^2} \sum_{i=1}^{n} \sum_{j=1}^{n} ||M_A^{i,j} - M_B^{i,j}||, \tag{4}$$

where $|| \cdot ||$ is the L2 norm function.

### 3.4 Overall Training Loss

Our overall training loss includes original adversarial training loss in StyleGAN2 and the similarity-based structure loss $L_{ss}$:

$$\begin{aligned} \mathcal{L}_G &= -E_{z \sim p(z)}[log(D(G(z)))] + \lambda_{ss}\mathcal{L}_{ss}, \\ \mathcal{L}_D &= -E_{z \sim p(z)}[log(1 - D(G(z)))] - E_{x \sim \mathcal{X}_t}[log(D(x))], \end{aligned} \tag{5}$$

where $\mathcal{X}_t$ is the training dataset. In our experiments, we set $\lambda_{ss} = 10$.

## 4 Experiments

We begin by introducing the experimental settings of our method, which encompass implementation, datasets, and metrics (Section 4.1). Next, we apply our method to various 10-shot datasets, showcasing its performance in Section 4.2. Section 4.3 explores the capabilities of domain association among multiple domains. Furthermore, we conduct ablation studies (Section 4.4 and Section A.1) to assess the impact of both the similarity-based structure loss and the generator structure. Additionally, we provide the user study in Section A.4 and demonstrate the applicability of our DoRM to one-shot GDA in Section A.3.

### 4.1 Experiments Settings

**Implementation.** We adopt the pre-trained StyleGAN2 [19] on FFHQ [18] as our source model, and our training parameters and settings follow StyleGAN2-ADA [17]. Due to the small amount of training data, we set the batch size to 4, and we terminate the training process after the discriminator has processed 100K real samples. Our implementation is based on the official implementation of StyleGAN2-ADA.

**Datasets.** Following previous literature [32; 59], we use FFHQ [18] with resolution $256 \times 256$ as the source domain. In 10-shot GDA, we evaluate our method on multiple target datasets, including Sketches [41], FFHQ-Babies [18], FFHQ-Sunglasses [18], Face-Caricatures, Face paintings by Amedeo Modigliani, Face paintings by Raphael, and Face paintings by Otto Dix [50]. All the training datasets are shown in Section A.2.

**Metrics.** Following [59; 32], we use Fréchet Inception Distance (FID) [13] as the metric to evaluate the synthesis quality and diversity simultaneously. Additionally, we adopt intra-cluster LPIPS (intra-LPIPS) [59; 32] based on LPIPS [54] to measure the synthesis diversity. Specifically, we synthesize 1000 images and then assign each of the synthesized images to the $k$ training images with the lowest

LPIPS distance, forming k clusters. We calculate the average LPIPS distance within each cluster and then average over all the clusters. Furthermore, we report the identity (ID) similarity [56] predicted by the Arcface [9] to measure the preservation of identity information, which is a metric to measure the cross-domain consistency.

## 4.2 10-shot Generative Domain Adaptation

**Qualitative comparison.** In 10-shot GDA, following the previous studies [32], we sample 10 training images from the target domain to transfer the pre-trained generator. Figure 3 shows the results of 10-shot GDA with different methods. We observe severe generator overfitting in the FreezeD [29]. GenDA [49] shows high quality and diverse synthesis but fails to maintain cross-domain consistency. CDC [32] and RSSA [47] improve the cross-domain consistency, but the synthesis quality is unsatisfactory. AdAM [58] also increases the synthesis diversity but fails in synthesis quality. By contrast, our method preserves all the domain-sharing attributes from the source domain by freezing the original generator and acquires the domain shift through new concurrent network components. Our method shows appealing synthesis quality and diversity while maintaining better cross-domain consistency than previous methods. More qualitative results can be found in Section A.2, Section A.5, and Section A.6.

**Quantitative comparison.** We use Fréchet Inception Distance (FID) to evaluate synthesis quality and diversity (lower is better). In 10-shot GDA, to better reflect the synthesis diversity , we use the Intra-LPIPS[32] to measure the synthesis diversity (higher means better synthesis diversity) . Additionally, we measure cross-domain consistency using Identify similarity (ID) [56] (higher is better). Table 1 summarizes the three evaluation metrics for 10-shot GDA over different target domains. FreezeD [29] and GenDA [49] struggle to maintain cross-domain consistency. CDC [32], RSSA [47], DCL [59] and AdAM [58] improve the synthesis diversity and cross-domain consistency but fails in the synthesis quality (as seen in Figure 3), leading to unsatisfactory FID scores. Our method outperforms all these methods on the different target domains, achieving not only better synthesis diversity and cross-domain consistency (higher Identity score and Intra-LPIPS) but also better synthesis quality (lower FID score).

Table 1: **Quantitative evaluation on 10-shot GDA**. All the source generators are pre-trained on FFHQ[18]. The target domains include Sketches, FFHQ-Baby and FFHQ-Sunglasses. Evaluation metrics include FID, Intra-LPIPS (I-LPIPS), and Identify similarity (ID). Noting that Sketches dataset only contains about 300 images, the large synthesis diversity will harm to the FID score.

| Datasets | FFHQ-Babies | | | FFHQ-Sunglasses | | | Sketches | | |
|---|---|---|---|---|---|---|---|---|---|
| Method | FID | I-LPIPS | ID | FID | I-LPIPS | ID | FID | I-LPIPS | ID |
| minegan | 98.23 | 0.514 | 0.132 | 68.91 | 0.42 | 0.171 | 64.34 | 0.40 | 0.092 |
| FreezeD | 110.92 | 0.346 | 0.037 | 51.29 | 0.337 | 0.030 | 46.54 | 0.325 | 0.010 |
| GenDA | 47.05 | 0.556 | 0.029 | 22.62 | 0.548 | 0.004 | **31.97** | 0.407 | 0.011 |
| CDC | 74.39 | 0.573 | 0.326 | 42.13 | 0.562 | 0.318 | 45.67 | 0.453 | 0.214 |
| RSSA | 77.77 | 0.576 | 0.314 | 70.41 | 0.563 | 0.307 | 63.44 | 0.480 | 0.296 |
| DCL | 52.56 | 0.582 | - | 38.01 | - | - | 37.90 | 0.486 | - |
| AdAM | 48.83 | 0.590 | 0.249 | 28.03 | 0.592 | 0.306 | 55.74 | 0.495 | 0.198 |
| **DoRM** | **30.31** | **0.623** | **0.445** | **17.31** | **0.644** | **0.389** | 40.05 | **0.502** | **0.365** |

## 4.3 Multi-domain and Hybrid-domain Generation

**Multi-domain generation.** Our proposed DoRM utilizes a novel approach by learning and retaining knowledge of new domains through the formation and update of new M&A modules, instead of updating the entire generator. This unique characteristic enables our DoRM, with a single generator, to generate images across multiple domains (as depicted in the top part of Figure 4). In contrast to previous methods [32; 47; 59; 58] that require updating the entire generator for few-shot GDA, limiting its capability into single domain generation, our DoRM offers significant storage space savings in multi-domain generation. As shown in Table 2, our method's models are about $3\times$ smaller than previous methods [32] on 10-domain generation.

**Hybrid-domain generation.** In contrast to previous methods [32; 59; 55] that fine-tune the entire generator, our DoRM achieves an effective linear domain shift through the update of new M&A

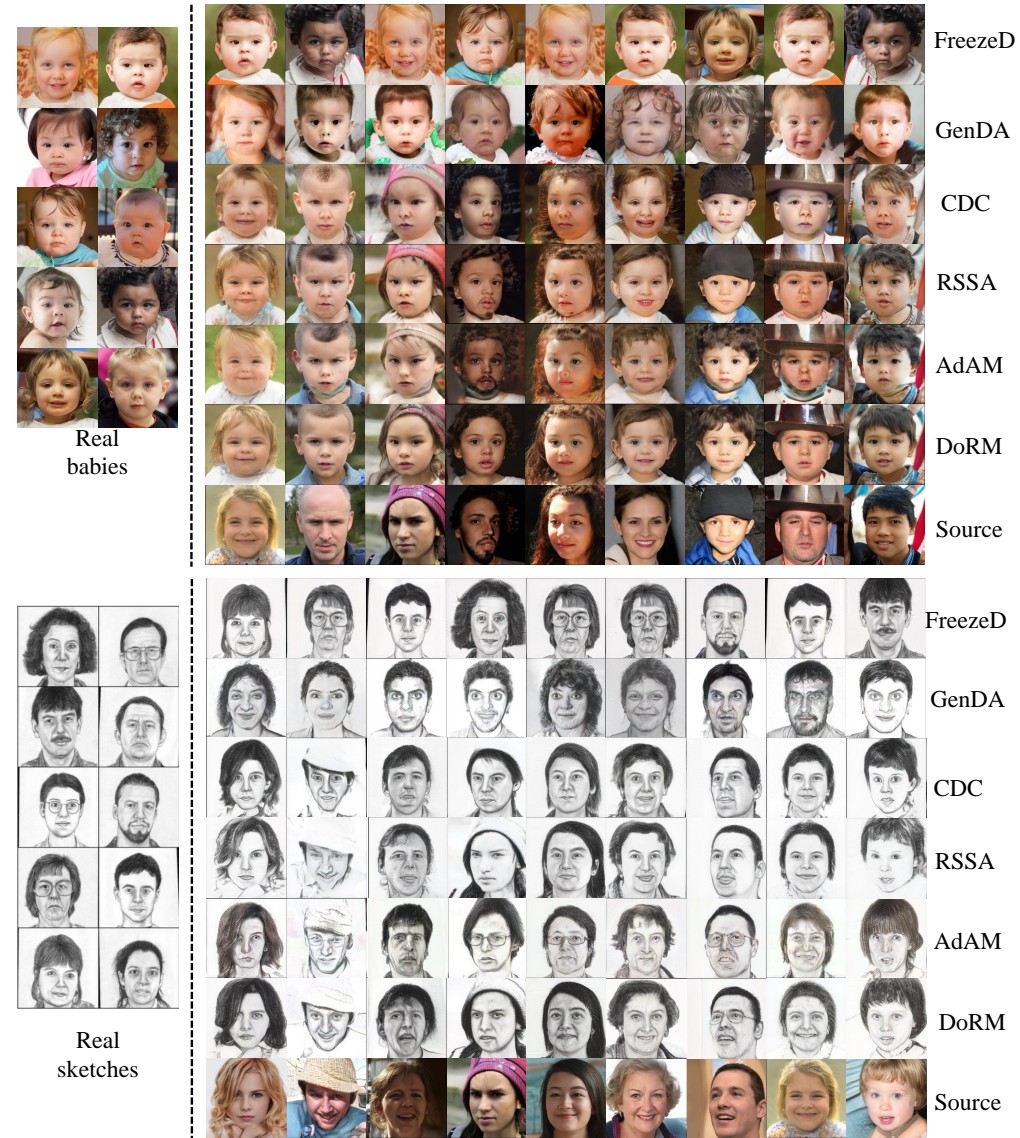

Figure 3: **Qualitative comparison on 10-shot GDA**. The source domain is FFHQ, and the target domains include Sketches and FFHQ-Babies. We compare our method with FreezeD[29], GenDA[49], CDC[32], RSSA[47] and AdAM[58]. Feeding the same latent code $z$ into the source generator and the target generator, we obtain the corresponding images in the source and target domains. Our method shows better cross-domain consistency and synthesis quality than the other methods.

modules. By learning multiple M&A modules that store the knowledge of various training domains, DoRM not only enables multi-domain generation but also facilitates the synthesis of images in hybrid domains not present in the training data (as depicted in the bottom left part of Figure 4). For comparison, we conduct hybrid-domain generation experiments using CDC [32]. In CDC, the domain shift between the source and target domain generators is obtained by subtracting their corresponding parameters. Similar to our DoRM, CDC achieves hybrid-domain generation by combining multiple domain shifts, adding the parameters of each shift to the source domain generator. However, as illustrated in Figure 4, CDC exhibits unsatisfactory synthesis quality in hybrid-domain generation. In contrast, the images synthesized by DoRM not only inherit domain-sharing attributes such as pose, gender, and identity, but also seamlessly blend domain-specific attributes. Furthermore, we extensively explore domain association among multiple domains using our proposed DoRM in Figure 5. The results demonstrate that DoRM efficiently integrates diverse domains to generate high-quality

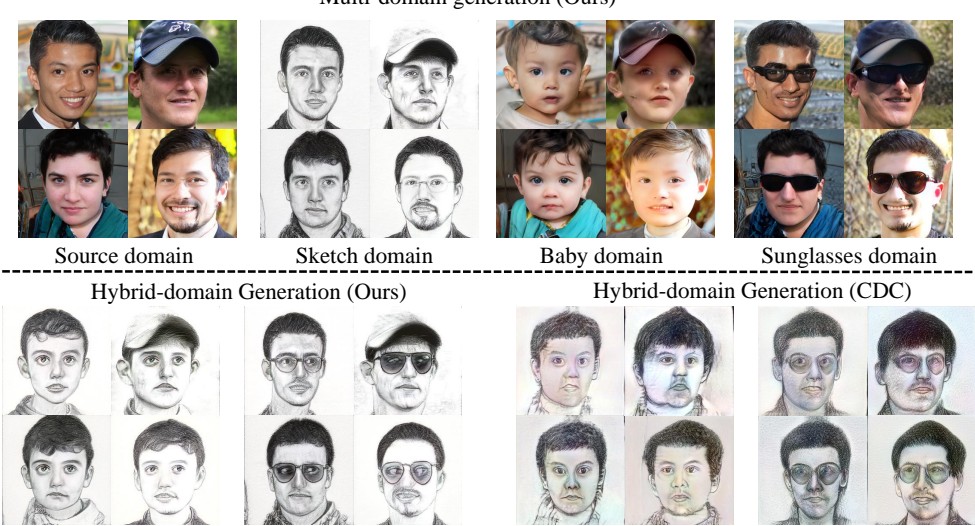

Figure 4: **Qualitative results on multi-domain and hybrid-domain generation.** The top part shows the multi-domain generation of our methods on three target domains. The bottom part illustrates the comparison on hybrid-domain generation between our proposed DoRM (bottom left) and CDC [32] (bottom right). The source generator is pre-trained on FFHQ [18]. The adopted M&A modules in our DoRM are trained in Section 4.2.

images in hybrid domains not present in the training data. In this case, our proposed DoRM has the ability to generate creative outputs in previously unseen domains by incorporating knowledge learned from other domains, similar to how humans can draw on past experiences to generate novel ideas. To the best of our knowledge, **our proposed DoRM is the first method to achieve efficient domain association in this manner**. Additional visualizations of hybrid domain generation can be found in Section A.6.

Table 2: **Storage Comparison.** The number of generator parameters required to realize different multi-domain generations.

| Model Size | 2-domain | 5-domain | 10-domain |
|---|---|---|---|
| CDC [32] | 48M | 120M | 240M |
| Ours | **30M** | **54M** | **84M** |

### 4.4 Ablation Study

We conduct ablation studies to evaluate the impact of our proposed DoRM generator structure and similarity-based structure loss. As shown in Figure 6, the images synthesized by baseline (which simply fine-tunes the StyleGAN2 generator with adversarial loss [18]) exhibit severe overfitting to the training images. By contrast, fine-tuning our DoRM structure generator with adversarial loss yields better synthesis diversity than the baseline. Moreover, when we optimize our DoRM generator structure with both adversarial loss and similarity-based structure loss, the synthesis images exhibit appealing quality and diversity, while maintain high cross-domain consistency, which indicates the proposed loss effectively preserves cross-domain consistency in few-shot GDA. More ablation studies on the network component of our DoRM are presented in Section A.1.

## 5 Limitation and Conclusion

**Limitation.** i) Although our method can achieve appealing results on the few-shot GDA across different target domains, however, the strength of domain shift $\alpha$ currently needs to be manually adjusted. Furthermore, since the various depth layers capture different semantic attributes, equipping each layer with an appropriate $\alpha$ may help to further improve synthesis quality. ii) The current

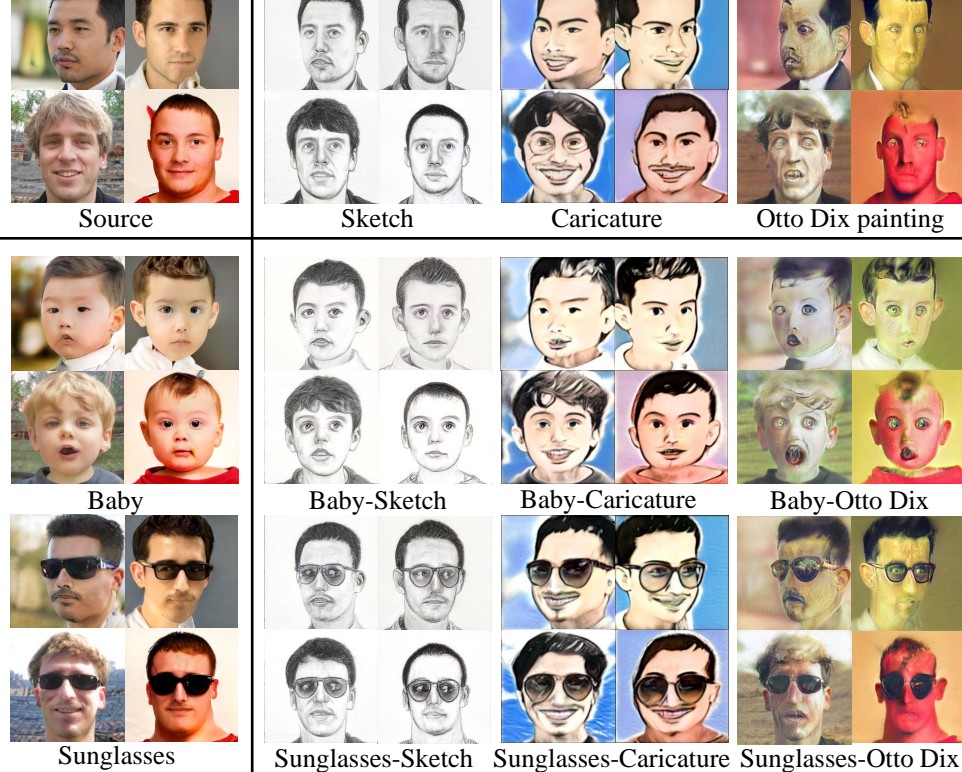

Figure 5: **Domain association between different domains by our DoRM.** The first row and the first column show the 10-shot GDA results on different target domains. By simply combining the corresponding pretrained M&A modules in 10-shot GDA, DoRM can integrate the domains and synthesize the high-quality images in the hybrid domains.

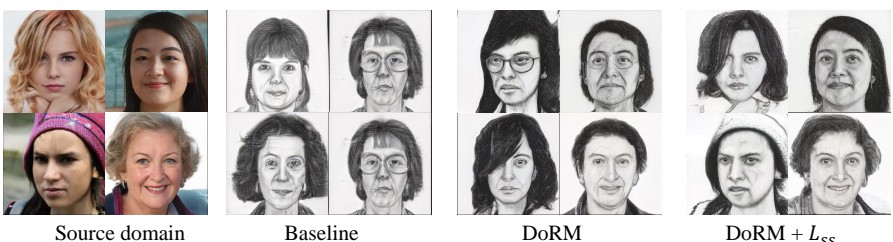

Figure 6: **Ablation study** on DoRM and $L_{ss}$. The training images are 10-shot sketches.

manuscripts only simply combines the M&A modules of different target domains and activate them at the same time to realize the domain association. To further improve the performance of the domain association, not only combining the trained target M&A modules but also employing a new M&A module and additional consistency loss is a better method to blend the target domains.

**Conclusion.** In this paper, we present DoRM, a novel generator structure inspired by the learning and storage mechanisms of the human brain, specifically designed for few-shot GDA. Our DoRM is characterized by its simplicity and efficiency. Additionally, we introduce a novel similarity-based structure loss to ensure cross-domain consistency in the few-shot GDA. Through extensive qualitative and quantitative evaluations, we demonstrate the superiority of our method over existing approaches in terms of synthesis quality, diversity, and cross-domain consistency. Importantly, akin to the human brain, our DoRM exhibits memory and the ability to integrate knowledge from different domains, enabling the generation of images in novel hybrid domains not encountered during training.

# 6   Acknowledgments

The work was supported in part by the National Natural Science Foundation of China under Grands U19B2044 and 61836011.

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

# A   Supplementary Materials

## A.1   Ablation Study

**Effect of Target Mapping.** In this section, we first investigate the significance of the target mapping module in our proposed DoRM. To evaluate the significance of the target mapping, we conduct an ablation study using the source mapping as a substitute for the target mapping, which remains frozen during the entire 10-shot generative domain adaptation training. As shown in Table 3, our results indicate a significant deterioration in the FID score without the target mapping, implying a considerable drop in the quality and diversity of the generated samples. Furthermore, Figure 7 illustrates that the generative domain adaptation barely occurs during training without the target mapping. This is because the target mapping plays a crucial role in capturing the representative attributes of the target domain and assisting in the acquisition of the domain shift during the adaptation process.

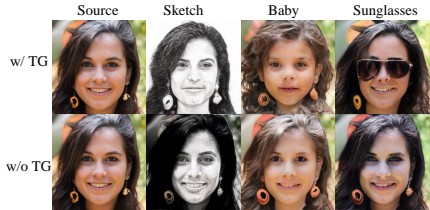

Figure 7: **Qualitative ablation study of the target mapping**. We compare the performance of our DoRM method, shown in the first row, with a variant of the method where the frozen source mapping is used as a substitute for the target mapping, depicted in the second row. We evaluate the performance of both methods on three different target domains: Sketches, FFHQ-Babies, and FFHQ-Sunglasses.

Table 3: **Quantitative ablation study of the target mapping**. The evaluation metric is FID (lower is better). We compare the performance of our DoRM method, shown in the first row, with a variant of the method where the frozen source mapping is used as a substitute for the target mapping, depicted in the second row. The source generator is pre-trained on FFHQ[18], and the target domains include FFHQ-Babies and FFHQ-Sunglasses.

|  | Babies | Sunglasses |
|---|---|---|
| DoRM | **30.31** | **17.31** |
| DoRM w/o Target Mapping | 86.52 | 74.71 |

**Effect of Re-Modulation Layers.** Another crucial component of our proposed DoRM approach is the target affine module. To investigate the roles of the different target affines in DoRM, we perform experiments where we drop the target affines in both the low-resolution and high-resolution feature maps. Specifically, we conduct 10-shot generative domain adaptation experiments to evaluate the Fréchet Inception Distance (FID) of the generated samples. For an image with a resolution of $256 \times 256$, the low-resolution feature maps include resolutions of $4 \times 4$, $8 \times 8$, $16 \times 16$, and $32 \times 32$. As presented in Table 4, our results demonstrate that all target affines are crucial for the performance of our DoRM approach and their removal leads to a significant drop in the quality and diversity of the generated samples.

**Effect of Re-Modulation Weight.** The re-modulation weight is a crucial parameter that controls the strength of the acquired domain shift in our proposed DoRM approach. A small re-modulation weight leads to a lower strength of the domain shift, resulting in more attributes of the source domain being preserved during generative domain adaptation. To investigate the impact of the re-modulation weight, we conduct 10-shot generative domain adaptation experiments using different re-modulation weights. The results are presented in Table 5. Our results demonstrate that different domain gaps have different optimal re-modulation weights, indicating that the selection of the re-modulation weight should be tailored to the specific target domain.

**Effect of Target domain classifier.** We investigate the effect of the target domain classifier on the performance of our proposed DoRM approach. Specifically, we experiment with different depths

Table 4: **Quantitative ablation study of the target affine layers**. The evaluation metric is FID (lower is better). We compare the FID score of our proposed DoRM method, shown in the first row, with two variants: one where the target affines are removed from the low-resolution feature maps, shown in the second row, and another where the target affines are removed from the high-resolution feature maps, shown in the third row. We use a source generator pre-trained on FFHQ [18] and evaluate all three methods on two different target domains: FFHQ-Babies and FFHQ-Sunglasses.

|  | babies | sunglasses |
|---|---|---|
| DoRM | **30.31** | **17.31** |
| DoRM w/o target affines in low resolution | 93.28 | 92.42 |
| DoRM w/o target affines in high resolution | 37.16 | 20.81 |

Table 5: **Quantitative ablation study of the re-modulation weight.** The evaluation metric is FID (lower is better). We conduct 10-shot generative domain adaptation experiments using our DoRM approach with varying re-modulation weights. We use a source generator pre-trained on FFHQ [18] and evaluate our method on two different target domains: FFHQ-Baby and FFHQ-Sunglasses.

| Re-modulation weight $\alpha$ | 0.5 | 0.2 | 0.05 | 0.005 | 0.001 |
|---|---|---|---|---|---|
| FFHQ-Baby | 37.9 | 36.1 | 34.0 | **30.3** | 32.3 |
| FFHQ-Sunglasses | 18.7 | **17.3** | 17.9 | 18.5 | 19.4 |

(*e.g.*, number of MLP layers) and initialization methods for the target domain classifier in 10-shot generative domain adaptation. As presented in Table 6, our results indicate that the two-layer MLP target domain classifier achieves the best performance. This is because the one-layer MLP lacks the ability to classify the target domain effectively, while the three-layer MLP is prone to overfitting due to the limited number of training images.

**Ablation of Individual Components of Previous State-of-the-art Methods.** To highlight the specific strengths of the proposed method that outperform other works, we consider the individual components and techniques utilized in previous state-of-the-art approach [55]. Specifically, we systematically analysis and compare individual components and techniques utilized in our DoRM and DiFa [55] to showcase how our method surpasses these works. According to the Introduction of the manuscript, GDA needs three fundamental properties. To resolve it, DiFa [55] proposed two CLIP-based loss: global loss $L_{global}$ and local loss $L_{local}$ for realizing large diversity/cross-domain consistency and high quality, respectively. Differently, we adopt Similarity-based Structure Loss ($L_{ss}$) and adversarial loss ($L_{adv}$) for realizing large diversity/cross-domain consistency and high quality, respectively. As shown in the Figure 8, the $L_{local}$ in DiFa mainly focuses on textual features and failes to capture the complete features (e.g. the white background in sketches) in generative domain adaptation. In our DoRM, we employ adversarial loss which can fully capture the features of the target images.

Table 6: **Quantitative ablation study of the target domain classifier**. The evaluation metric is FID (lower is better). We experiment with different depths of the target domain classifier, using various initialization methods. We use a source generator pre-trained on FFHQ [18] and evaluate our proposed DoRM approach on 10-shot generative domain adaptation tasks.

| MLP Depth | FFHQ-Baby | | FFHQ-Sunglasses | |
|---|---|---|---|---|
|  | source initial | random initial | source initial | random initial |
| one layer | 38.03 | 37.15 | 24.63 | 22.15 |
| two layers | 33.32 | **30.31** | 20.40 | **17.31** |
| three layers | — | 33.25 | — | 20.42 |

## A.2 More Synthesis Results in 10-shot GDA

**Qualitative results on 10-shot generative domain adaptation.** We present qualitative results of our proposed DoRM approach on more target datasets, including face caricatures, face paintings by Raphael, face paintings by Amedeo Modigliani, and face paintings by Otto Dix [50]. All training images are shown in Figure 9, and the qualitative results are presented in Figure 10. Our results

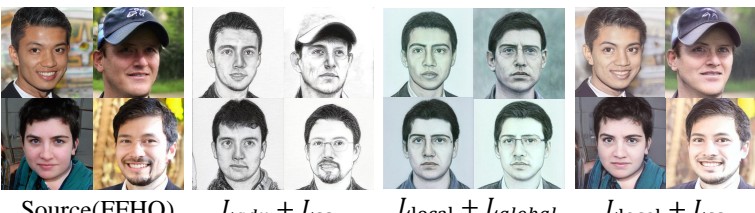

| Source(FFHQ) | $L_{adv} + L_{ss}$ | $L_{\text{local}} + L_{global}$ | $L_{\text{local}} + L_{ss}$ |

Figure 8: Ablation of individual components of previous state-of-the-art methods.

demonstrate that our proposed DoRM approach achieves appealing synthesis quality in various target domains.

**Results of latent interpolation.** We perform latent space interpolation to demonstrate that our DoRM is not harmful to the learned latent space. In Figure 11, the first and last columns show the generated images with two latent codes after 10-shot generative domain adaptation, while the remaining columns show the results obtained by linearly interpolating the two latent codes. Our results demonstrate that all intermediate synthesized images have high target-domain consistency and high cross-domain consistency. Moreover, the semantics of the generated images, such as gender, haircut, and pose, vary gradually throughout the interpolation, indicating that our proposed DoRM approach preserves the underlying semantic structure of the learned latent space.

**Results of latent editability.** We have performed editing on a real image adapted into a new target domain using StyleGAN-CLIP to discover editing directions in the source domain. The results, as illustrated in Fig 12, indicate that the adapted generator maintains similar latent-based editing capabilities to the original generator. This demonstrates the preservation of editability in the adapted generator.

**Results on 3D generator.** We conducted some initial studies using the popular 3D-aware image generation method, EG3D [3], for one-shot GDA with FFHQ as the source domain and Sketch as the target domain. The results, as shown in the Figure 13, reveal that directly migrating the proposed DoRM to 3D one-shot domain adaptation poses challenges and might not be straightforward. Some potential challenges in the 3D-GAN domain include:

(1) Overfitting: 3D GANs often face more severe overfitting issues when dealing with limited training data, requiring more rigorous regularization techniques and training strategies for one-shot GDA.

(2) Volumetric Data Representation: Handling volumetric data representation in 3D-GANs necessitates specialized techniques for data manipulation, augmentation, and visualization.

(3) Spatial Artifacts: Generating high-quality 3D objects may encounter spatial artifacts, such as geometric distortions or inconsistent shapes, which need to be addressed.

(4) Computational Complexity: The computational demands of 3D-GANs can be significantly higher than their 2D counterparts, presenting challenges in both training and inference.

**Results on three-domain hybrid domain generation.** As depicted in the Figure 14, we have showcased an example of applying our method to create hybrid-domain images by activating the trained M&A modules of Baby, Sunglasses, and Sketch domains. This illustration demonstrates how easily our approach can be adapted to generate hybrids involving more than two domains, showcasing the versatility and potential of our approach.

### A.3 Experiments on one-shot Generative Domain Adaptation

Although our DoRM is mainly for few-shot generative domain adaptation, DoRM is also can be employed for one-shot generative domain adaptation [21; 55; 6]. In the one-shot GDA, the training dataset is a single image, which is difficult for the backbone of discriminator to extract the main characters of the target domain because of the overfitting issue. In this case, we introduce a clip-based local-level adaptation loss $L_{local}$ from [55] to help to acquire the local-level characters and styles of the target domain. Concretely, we extract the intermediate tokens of the adapted image $I_B$ synthesized by DoRM and the single target image $I_{tar}$ from the $k - th$ layer of CLIP image encoder. And align each of adapted token $F_B$ with its closest target token from $F_{tar}$, where $F_B = F_B^1, ..., F_B^n$ and

$F_{tar} = F_{tar}^1, ...F_{tar}^m$ are the extracted tokens. The clip-based local-level adaptation loss is defined as:

$$L_{local} = \max(\frac{1}{n}\sum_i \min_j C_{i,j}, \frac{1}{m}\sum_j \min_i C_{i,j}) \tag{6}$$

where $C$ is calculated as:

$$C_{i,j} = 1 - \frac{F_B^i \cdot F_{tar}^j}{|F_B^i||F_{tar}^j|} \tag{7}$$

Furthermore, to better identify and maintain the domain-sharing attributes in one-shot generative domain adaptation, we also employ the inversion-based selective cross-modal consistency loss $L_{scc}$ from [55]. Specifically, this loss function aims to identify and preserve domain-sharing attributes in the $W+$ space. The underlying assumption is that attributes that are similar in $W+$ space between the source and target domains during adaptation are more likely to be domain-sharing attributes. To achieve this, $L_{scc}$ dynamically analyzes and retains these attributes. First, it inverts the source and corresponding target images into $W+$ latent codes, $w_A$ and $w_B$, respectively, using a pre-trained inversion model such as pSp pr e4e, for each iteration. Next, it computes the difference $\Delta w$, between the centers of a source queue of $W+$ latent codes, $X_A$ and the target queue of $W+$ latent codes, $X_B$, where $X_A$ and $X_B$ are dynamically updated with $w_A$ and $w_B$ during training. The loss function then encourages $w_A$ and $w_B$ to be consistent in channels with less difference, thereby facilitating the preservation of domain-sharing attributes. The inversion-based selective cross-modal consistency loss $L_{scc}$ is defined as follows:

$$L_{scc} = ||mask(\Delta w, \alpha) \cdot (w_B - w_A)||_1 \tag{8}$$

where $\alpha$ represents the proportion of preserved attributes, and $mask(\Delta w, \alpha)$ determines which channels to retain. Specifically, let $|\Delta w_{s_{\alpha N}}|$ be the $\alpha N - th$ largest element of $\Delta w$. Then, each dimension of $mask(\Delta w, \alpha)$ is calculated as follows:

$$mask(\Delta w, \alpha)_i = \begin{cases} 1 & |\Delta w_i| \leq |\Delta w_{s_{\alpha N}}| \\ 0 & |\Delta w_i| \geq |\Delta w_{s_{\alpha N}}| \end{cases} \tag{9}$$

We compare our DoRM++ approach which denotes introducing the two new loss terms into training with state-of-the-art one-shot generative domain adaptation (GDA) methods, including JoJoGAN [6], Generalized One-shot Domain Adaptation [56], DynaGAN[20] and DiFa [55]. Figure 16 shows the comparison results. Our results indicate that JoJoGAN, DynaGAN and Generalized One-shot Domain Adaptation fail to achieve GDA when the target image is FFHQ-Baby and FFHQ-Sunglasses, and the synthesis quality of DynaGAN is limited. Similarly, DiFa also fails to achieve GDA when the target image is FFHQ-Sunglasses, and the synthesis diversity is unsatisfactory when the target image is FFHQ-Baby.

In contrast, our DoRM++ approach achieves one-shot GDA among all the reference images, resulting in high-quality and diverse synthesis, while maintaining appealing cross-domain consistency. Moreover, our DoRM++ generator has memory to realize multiple target domains' generation, which saves a significant amount of storage space. Our DoRM++ generator also has the ability to integrate the learned knowledge of multiple target domains to synthesize images in hybrid domains that are unseen in the target domains. As shown in Figure 17, the DoRM++ generator can synthesize high-quality and diverse images in hybrid domains while maintaining the domain-sharing attributes (e.g, pose, identity).

### A.4 User Study in One-shot GDA

In this section, we have planned to conduct a user study in one-shot GDA to enhance our evaluation process. Specifically, our user study involves presenting users with a reference image, a source image, and four adapted images from different methods (DORM++, Generalized One-shot Domain Adaption [56], DynaGAN [20], and DiFa [55]). We will ask users to choose the best adapted image for each of three measurements: (i) image quality, (ii) style similarity with the reference, and (iii) attribute consistency with the source image. To ensure statistical significance, we will generate 500 samples for each method and involve 50 users in our study. Each user will be randomly assigned 50 samples from the 500, and they will have unlimited time to complete the evaluation. As shown in Table 7, preliminary results indicate that users strongly favor our DoRM in all three aspects, reflecting the effectiveness of our approach compared to the alternative methods.

Table 7: User study on one-shot GDA. The numbers represent the percentage of users who favor the images synthesized corresponding method among the all four methods.

| Model Comparison | image quality | style similarity | attribute consistency |
|---|---|---|---|
| DoRM++ (Ours) | 59.02% | 69.35% | 67.16% |
| Generalized One-shot Domain Adaption [56] | 7.38% | 16.33% | 11.79% |
| DynaGAN [20] | 1.44% | 3.73% | 3.41% |
| DiFa [55] | 32.16% | 10.59% | 18.24% |

## A.5 Experiments of DoRM and DoRM++ on the one-shot and 10-shot GDA

Figure 15 thoroughly explore the performance of DoRM++ in a 10-shot GDA scenario and that of DoRM in a one-shot GDA context. The results of these experiments illustrate that both DoRM++ and DoRM exhibit strong performance in both few-shot and one-shot GDA scenarios. Notably, DoRM++ showcases enhanced cross-domain consistency compared to DoRM in the context of one-shot GDA.

## A.6 Hybrid-domain Generation

In Figure 17, we present the results of generating hybrid domains using our proposed model. Our DoRM has a unique generator structure that is similar to the mechanism of the human brain. This structure endows the DoRM with two novel capabilities: memory and domain association. These capabilities enable the DoRM to not only retain knowledge from previously learned domains when generating images in new domains, but also integrate multiple learned domains and synthesize images in hybrid domains that were not encountered during training.

Additionally, we proceed to provide a comparative analysis of three methods within this section: Our DoRM/DORM++, DynaGAN [20], and Domain Expansion [30]. As illustrated in Figure 18, we strive to alleviate any confusion by conducting comprehensive experiments on the memory and domain association capabilities of these three methods under one-shot and 10-shot settings. These results effectively demonstrate that the baseline methods exhibit poorer performance across various settings compared to our DoRM/DoRM++. Notably, both DynaGAN and Domain Expansion experience difficulties in achieving successful hybrid domain generation, highlighting the distinctive advantages of our approach.

Finally, we meticulously scrutinized the quantitative experiments conducted within the Domain Expansion framework, leading us to adopt a cosine similarity evaluation metric termed "domain similarity" ($Sim$) based on CLIP image encoder ($E_I$). This metric is employed to provide a quantitative assessment of the fidelity exhibited by hybrid domains. In detail, for given generative images in the hybrid domain "sketch-baby" ($I_{SB}$), we extract image features from the provided images ($E_I(I_{SB})$) as well as its corresponding target images ($E_I(I_S)$ and $E_I(I_B)$). Therefore, the "domain similarity" to "sktech" and "baby" domain are defined as $Sim1 = cos(\overline{E_I(I_{SB})}, \overline{E_I(I_S)})$ and $Sim2 = cos(\overline{E_I(I_{SB})}, \overline{E_I(I_B)})$, respectively.

Subsequently, we compute the cosine similarity for each case. The quantification of results from both the one-shot and 10-shot experiments is meticulously presented in Table 8 and Table 9, respectively. These outcomes distinctly illustrate the remarkable performance of our proposed DoRM in both single-domain and hybrid-domain generation. It is imperative to highlight a notable observation amidst these findings: a certain outlier exists. Specifically, the images generated by DynaGAN and HyperdomainNet within the "elsa-sunglasses" domain exhibit a remarkably high domain similarity with the "sunglasses" domain, while conversely displaying significantly lower domain similarity with the "elsa" domain. This phenomenon can be rationalized by the generated images closely resembling the "FFHQ" domain, as depicted in Figure 6 of the attached PDF in the rebuttal. Consequently, the domain similarity to the "FFHQ-sunglasses" domain also emerges as notably high. Drawing upon this observation, we emphasize the necessity for a comprehensive evaluation of hybrid-domain generation that seamlessly integrates both qualitative and quantitative results.

## A.7 Experiments on Other Source Domains

In addition to the experiments on FFHQ, we conduct other 10-shot generative domain adaptation experiments to qualitatively evaluate the effectiveness of our proposed DoRM approach. Specifically,

we pre-trained a StyleGAN2 on the LSUN-church [52] dataset and adapted the pre-trained GAN to generate haunted house images. The results of our experiments are presented in Figure 19.

Table 8: Cosine similarity (higher is better) of CLIP feature between generated images and corresponding target images on one-shot experiments.

| Method | elsa | baby | sunglasses | elsa-baby | elsa-sunglasses |
|---|---|---|---|---|---|
| DynaGAN | 0.9165 | 0.7866 | 0.8882 | 0.6196/0.7832 | 0.6245/0.8467 |
| DynaGAN-interpolation | - | - | - | 0.6201/0.7836 | 0.6330/0.8546 |
| HyperdomainNet | 0.8740 | 0.7739 | 0.8589 | 0.7007/0.7041 | 0.6554/0.7882 |
| Domain expansion | 0.9075 | 0.9614 | 0.9339 | 0.7022/0.7762 | 0.7671/0.6177 |
| Ours | 0.9309 | 0.9814 | 0.9370 | 0.7173/0.7842 | 0.7734/0.6377 |

Table 9: Cosine similarity (higher is better) of CLIP feature between generated images and corresponding target images on 10-shot experiments.

| Method | Sketches | Babies | Sunglasses | Sketches-Babies | Sketches-Sunglasses |
|---|---|---|---|---|---|
| Domain expansion | 0.8958 | 0.9546 | 0.9094 | 0.7480/0.7112 | 0.7720/0.6735 |
| Ours | 0.9492 | 0.9780 | 0.9136 | 0.8809/0.7271 | 0.8057/0.6973 |

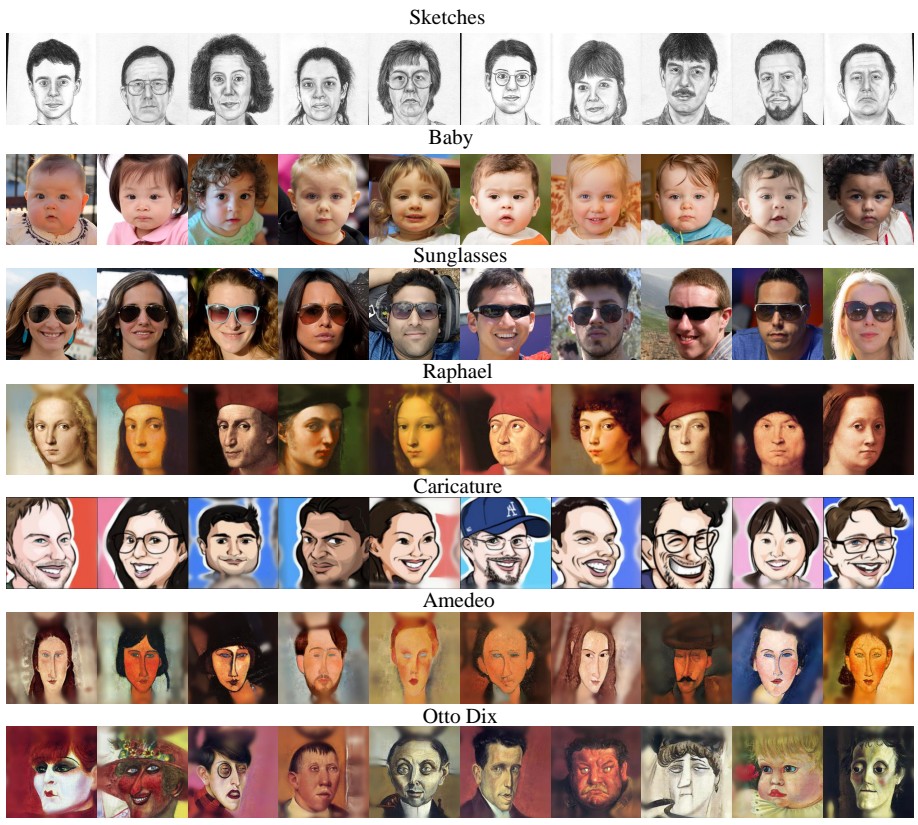

Figure 9: **Training images in 10-shot generative domain adaptation experiments**.

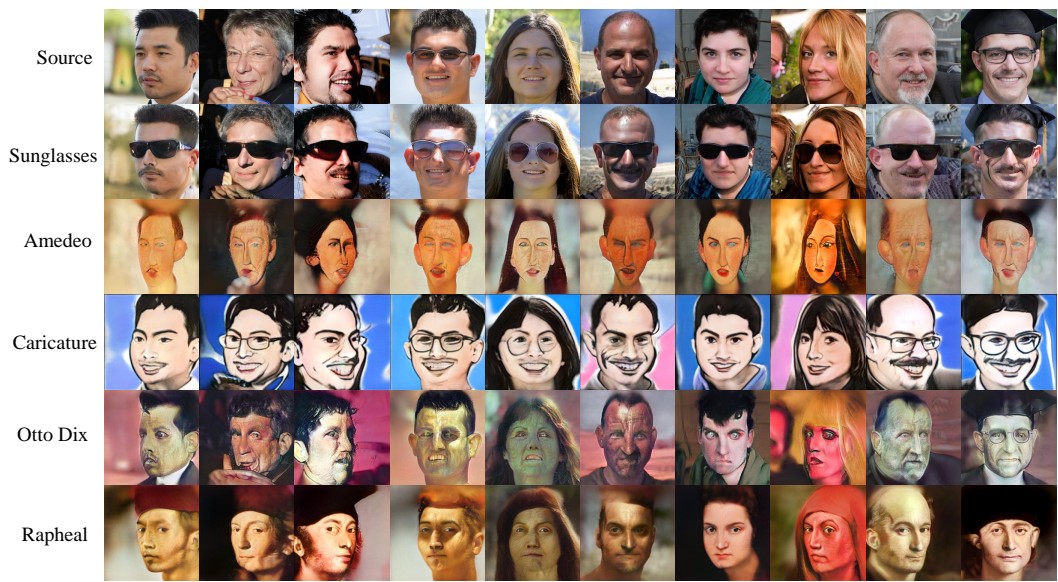

Figure 10: **10-shot generative domain Adaptation on FFHQ**. We use a source generator pre-trained on the FFHQ [18] dataset and evaluate our proposed DoRM approach on various target domains, including FFHQ-Sunglasses, face caricatures, face paintings by Raphael, face paintings by Amedeo Modigliani, and face paintings by Otto Dix [50]. The training images are shown in Figure 9. Our results demonstrate that our proposed DoRM approach can maintain cross-domain consistency between the source domain and different target domains.

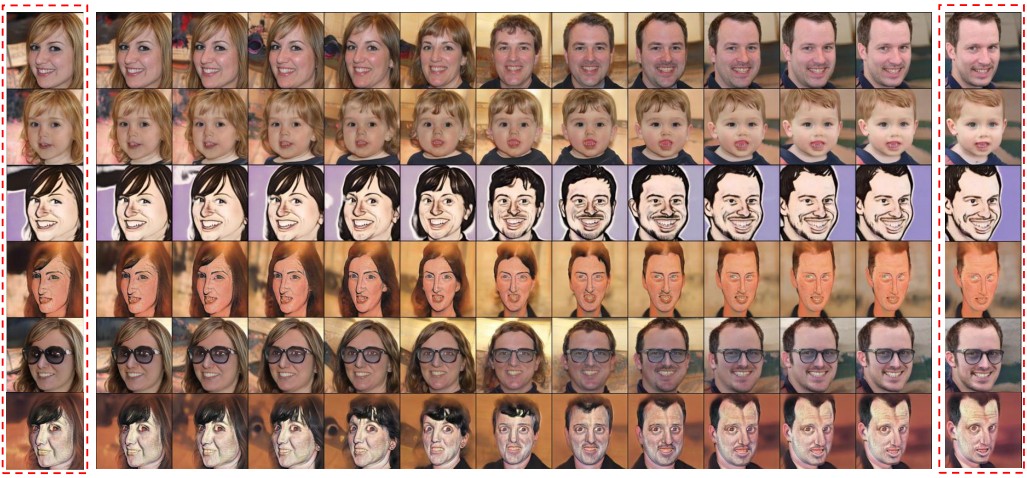

Figure 11: **Latent interpolation** using the generators adapted to different target domains in 10-shot generative domain adaptation (the first line is the source images). The first and last columns are the generated images with two latent codes after 10-shot generative domain adaptation. The remaining columns are the results by linearly interpolating the two latent codes. According to the figure, all the semantics (*e.g.*, the gender, the haircut and the pose) vary gradually.

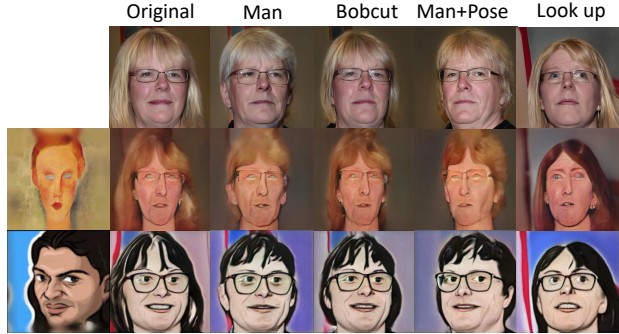

Figure 12: **Latent edit** using the generators adapted to different target domains in 10-shot generative domain adaptation (the first line is the source images). The second and third lines are two popular target domain images, where the first column is the example of 10-shot target images and the other columns are adaptation results.

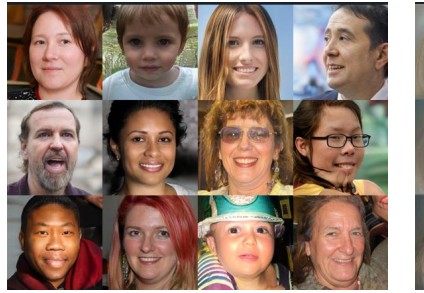 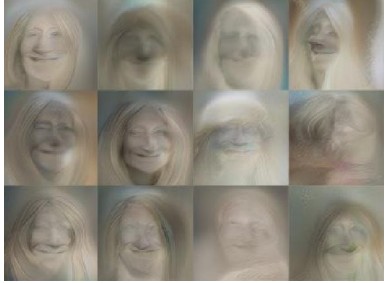

Figure 13: **One-shot GDA in 3D GANs.** DoRM for one-shot GDA in EG3D. Left: source image in FFHQ. Right: the generative target domain (Sketch) images.

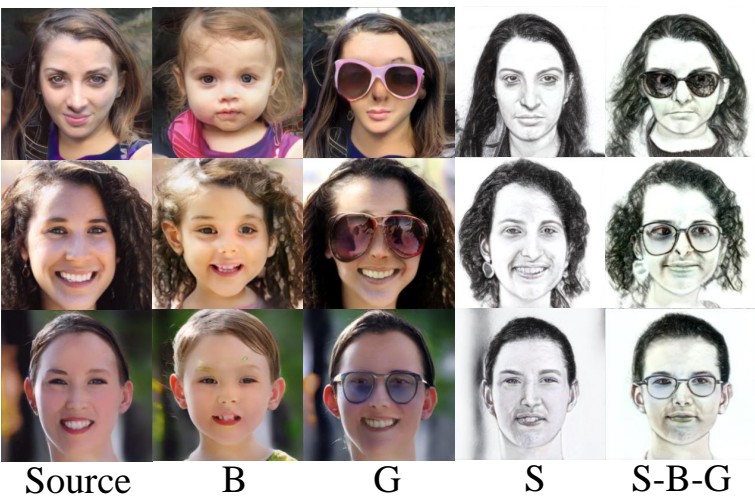

Figure 14: Three-domain hybrids generation on 10-shot Babies (B), Sunglasses (G), and Sketch (S) datasets. This illustration demonstrates how easily our approach can be adapted to generate hybrids involving more than two domains.

**one-shot**

**10-shot**

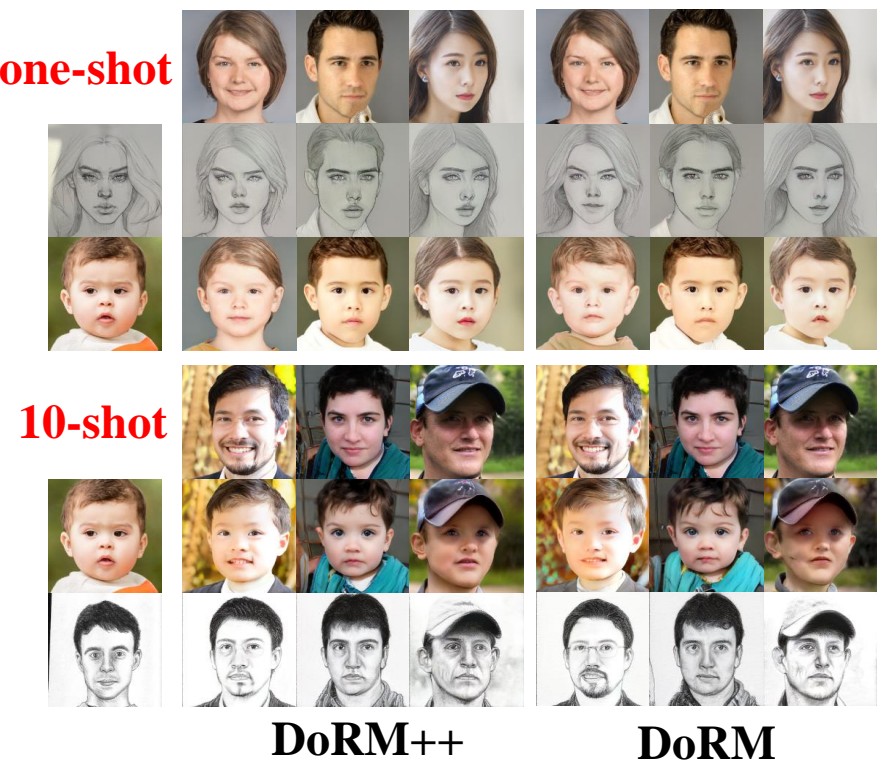

**DoRM++**          **DoRM**

Figure 15: **Results of DoRM and DoRM++ on the one-shot and 10-shot GDA**.

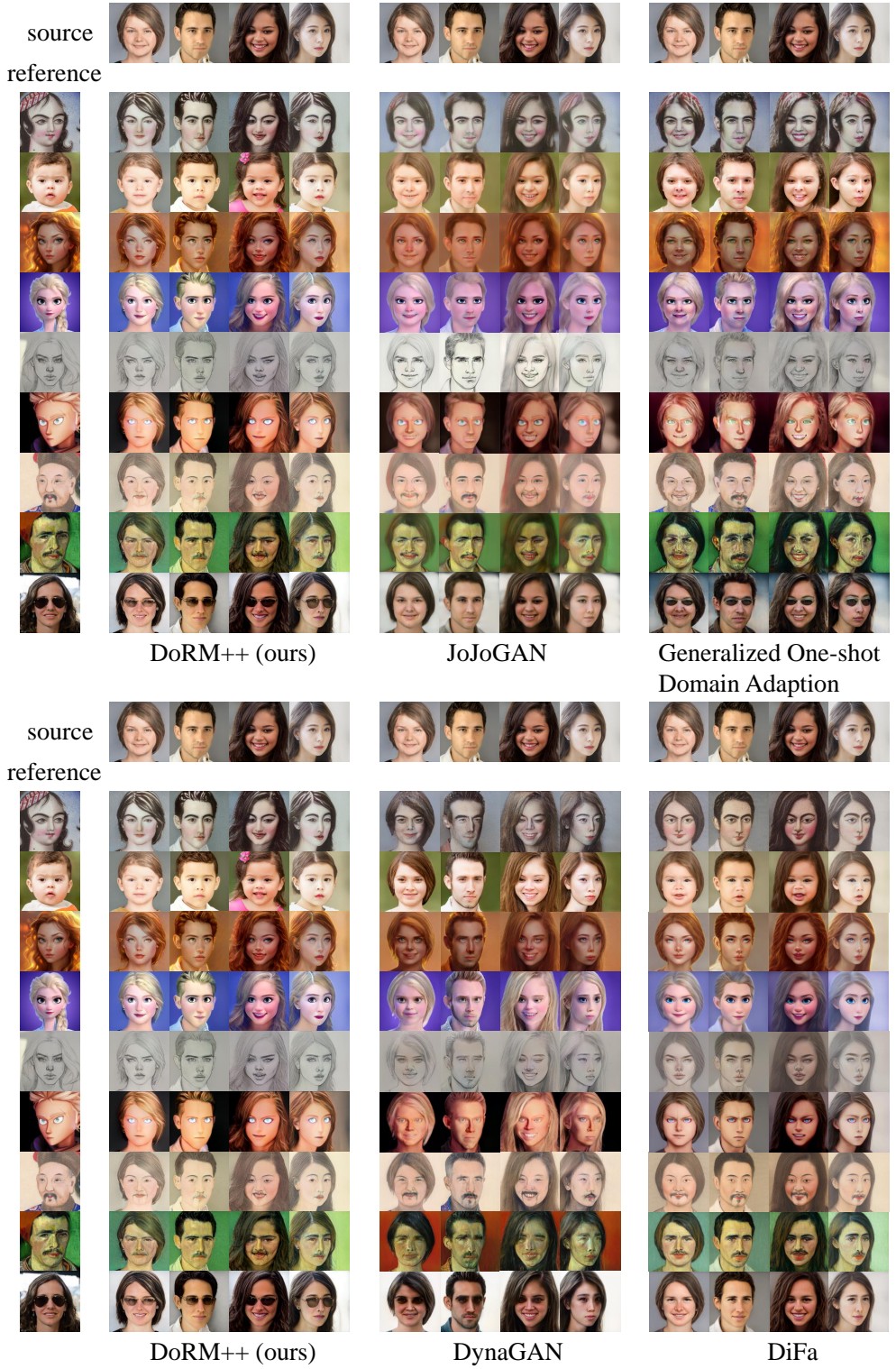

Figure 16: **Qualitative comparison on one-shot GDA**. The source domain is FFHQ, and the target domains include different reference images, as shown in the first row of the figure. We compare our method with JoJoGAN [6], Generalized One-shot Domain Adaption [56], DynaGAN[20] and DiFa[55]. Our method not only achieves better synthesis quality and diversity but also maintains higher cross-domain consistency than other methods.

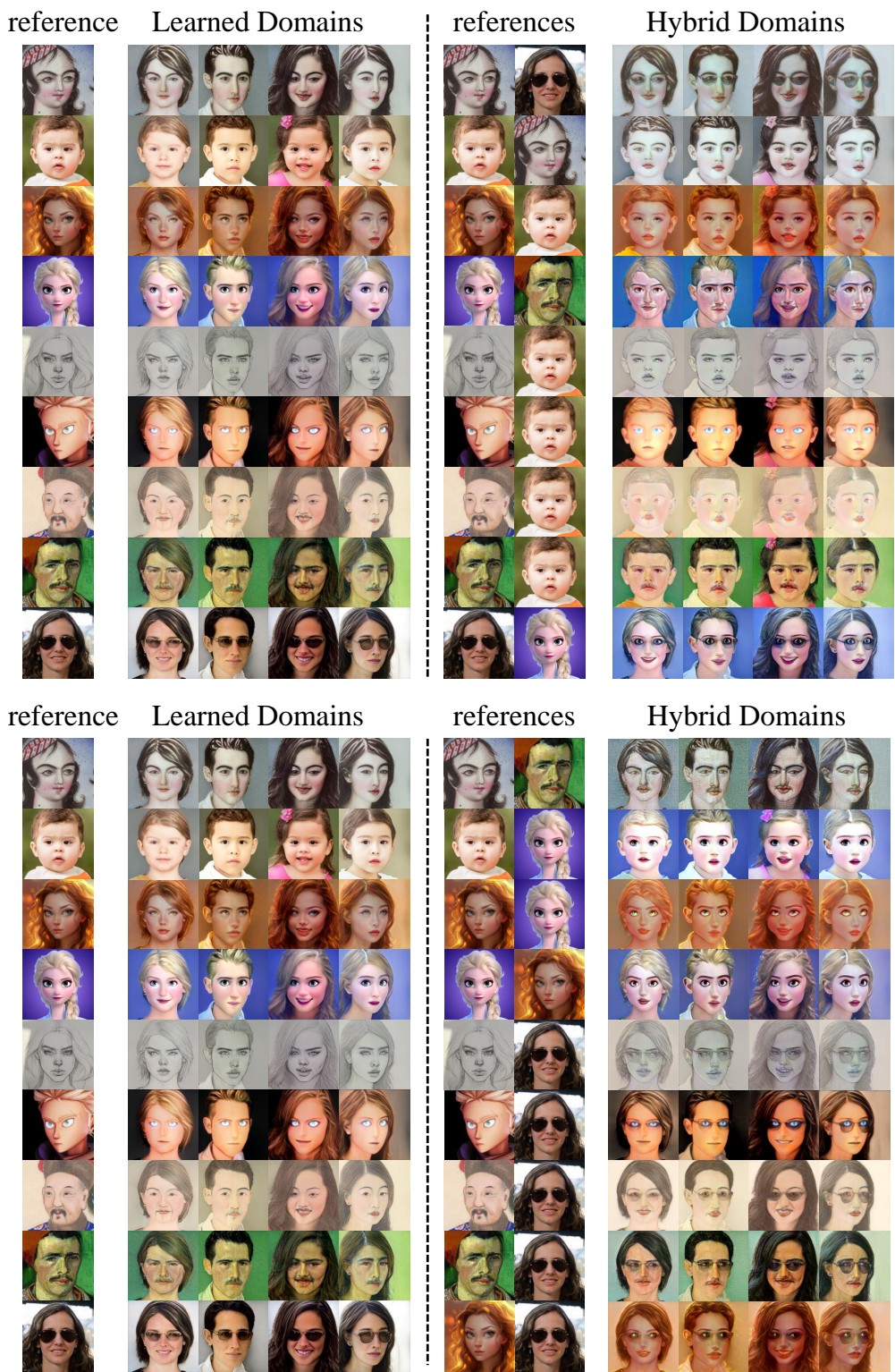

Figure 17: **One-shot generative domain adaptation and domain association on FFHQ**. The source domain is FFHQ, and the target domains include different reference images, as shown in the first column of the figure. Once our DoRM++ generator learns to synthesize images in multiple target domains, it can integrate the knowledge from the learned multiple domains and synthesize images in hybrid domains which are unseen in the target domains.

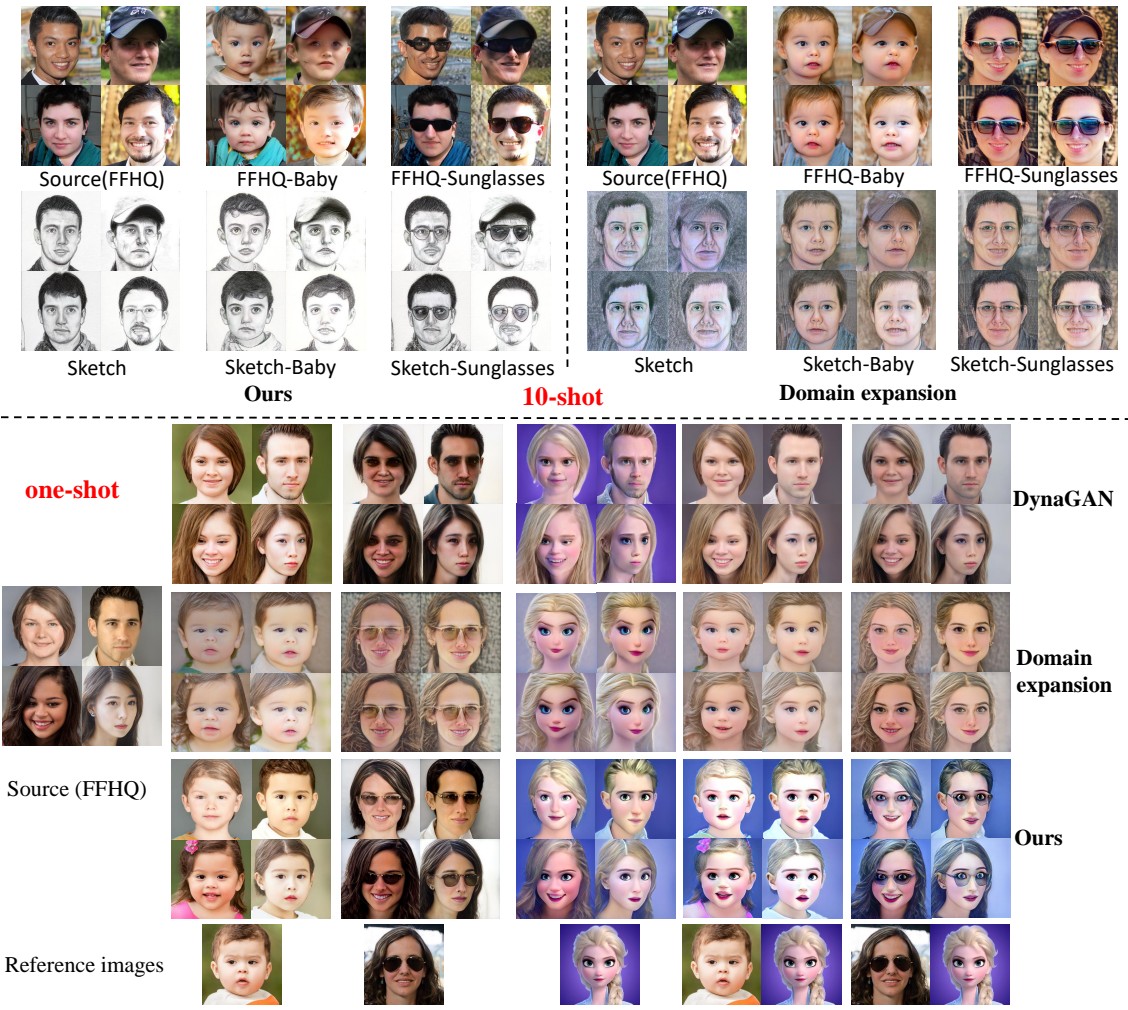

Figure 18: Comparison with baselines on 1-shot and 10-shot GDA. The outcomes underscore our DoRM's superiority, revealing inferior performance by baseline methods across various scenarios. Remarkably, other methods face challenges in achieving successful hybrid domain generation.

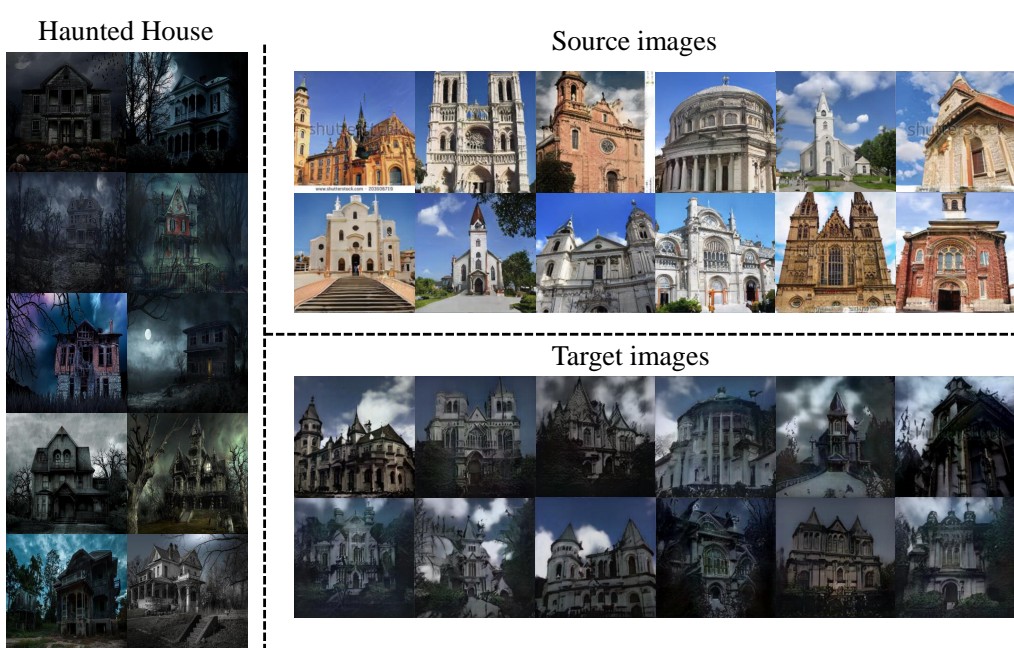

Figure 19: **10-shot generative domain adaptation on LSUN-Church[52]**. The source generator is pretrained on LSUN-Church[52]. The target domain is haunted house (10 training images are shown on the left side). The result shows that our method can maintain the cross-domain consistency between the source domain and the target domain.

