# OpenReview forum: "Domain Re-Modulation for Few-Shot Generative Domain Adaptation"
_NeurIPS.cc/2023/Conference — NeurIPS 2023 poster_

### Official Review · Reviewer_URf4 · 2023-06-30

**Soundness:** 1 poor
**Presentation:** 2 fair
**Contribution:** 2 fair
**Rating:** 3
**Confidence:** 5

**Summary:**

This work studies generative domain adaptation (GDA), on the StyleGAN2 architecture. Methods in this setting are commonly evaluated in terms of quality, diversity, and cross-domain consistency. This work claims to also be the first to explore the abilities of memory (“retain knowledge from previously learned domains”) and domain association (“integrate multiple learned domains and synthesize hybrid domains”).  To tackle all these aspects, a method called DoRM is proposed. DoRM learns a new mapping network and affine transformations – M&A - (components of StyleGAN2) for each new domain. Hybrid domains are generated by interpolating the style codes of the original mapping and affine layers with the new ones.

**Strengths:**

1.	The paper is clearly written and easy to follow.
2.	The tasks studied in this paper are interesting and relevant.
3.	The proposed method is intuitive and novel, and as demonstrated by some experiments it is effective in performing domain adaptation.


**Weaknesses:**

The claimed contributions are incorrect, and the evaluation is severely lacking. Both issues stem from outright ignoring the role and contributions of the most relevant previous works – HyperDomainNet [1], DynaGAN [18], Domain Expansion [24]. Below I list several examples of severe problems with claims and evaluations.

All three works consider settings that cover the supposably new aspects of domain adaptation that are discussed in this paper. Both DynaGAN and HyperDomainNet rely on similar modulation techniques and achieve *”memory”* the same way. In Domain Expansion, the original domain is preserved on a dedicated subspace. Also, both DynaGAN and Domain Expansion discuss *”association”* in length (using the terms “domain interpolation” and “domain composition”). So, clearly, this work’s claims (lines 64, 247) of being the first to consider this setting are false.

The previous fact is even somewhat acknowledged in the Related Work section (lines 99-105). The three works are said to “fall short in integrating multiple domains”. However, this claim is not supported by any experiment.
Not only this one claim is not supported, but the authors choose to not compare their method with the three most relevant works - all of which have code on Github and all have been published well before the submission deadline (October 2022-January 2023). Instead, the method is compared to the more conventional setting of “single-domain, entire generator” adaptation, which is negligent at best. Comparing the generation of “hybrid” domains to CDC [25] which was not designed for this purpose (and published in 2021) and not to DynaGAN and Domain Expansion is unreasonable. Similarly, claiming superiority over CDC in terms of storage required is irrelevant. DoRM trains a mapping network (~6M params) for each new domain, while HyperDomainNet trains ~6K params, and Domain Expansion requires no additional weights whatsoever!

The inaccurate contribution claims and lack of comparison to relevant previous works is fatal as it prevents the community and practitioners from understanding the landscape of works in this field. Perhaps the paper was written around the end of 2022 and is simply not up to date. I urge the authors to consider significantly rewriting this paper before resubmitting or arXiving a preprint.

Additional issues:
1. Measuring consistency between domains using ArcFace – a face recognition network trained on faces -- makes little sense to me. For example, why would it output anything meaningful on sketches?
2. Missing citation to StyleSpace [Wu et al.] who introduced the $\mathcal{S}$ used in this paper (line 120).


**Questions:**

1.	Please clarify in what way is the setting that is supposably suggested was not already covered by HyperDomainNet, DynaGAN and Domain Expansion?
2.	Why did you not include comparisons to these methods despite acknowledging that they are highly similar and relevant?


**Limitations:**

One limitation is mentioned. However, several questions remain:
1. Can hybrids be produced by combining more than two domains?
2. Are the produced hybrids less faithful to each individual domain and to what extent?

---

> ### Author Rebuttal · Authors · 2023-08-09
>
> **Q1. The claimed contributions are incorrect, and the evaluation is severely lacking**
>
> We apologize for any oversight and plan to enhance the quality of our paper through the following steps:
>
> **1. Thorough Comparison and Differentiation from Prior Works**. It has demonstrated in the [General Response: Comprehensive Comparison and Discernment from Pertinent Works [1] [2] [3]](https://openreview.net/forum?id=jown9RvYn7&noteId=xNZlQlaQNI)
>
> **2. Clarification of First-to-Consider Claim.** We acknowledge that our manuscript has inaccurately represented our major contributions. In Section 2.1 of the manuscript, we have indeed recognized that the three mentioned  methods [1][2][3] achieve comparable domain memory capabilities to our approach. However, we find that both HyperDomainNet [1] and DynaGAN [2] lack the capacity to realize "Domain Association" due to their reliance on scale modulation parameters in the StyleGAN style space. Additionally, these methods did not thoroughly explore the potential applications of "Domain Association" in their respective papers. It is essential to clarify that the "domain interpolation" technique in DynaGAN [2] significantly differs from our concept of "Domain Association." While "domain interpolation" facilitates a seamless transition between two target domains through vector interpolation, it falls short in creating a hybrid domain that simultaneously embodies the properties of multiple fundamental domains, as seen in our "sketch"+"Sunglasses" domain example. We acknowledge the "domain composition" aspect discussed in [3] aligns with our proposed "Domain Association." Regrettably, our initial review of the references was incomplete, and we overlooked the significance of [3]'s contribution to the domain association. To rectify this, we will revise the description of our main contributions and provide a more comprehensive analysis of related works.
>
> **3. Comprehensive Evaluation of Methods.** In light of our paper's focus on "few-shot GDA", we conducted qualitative and quantitative comparisons using established criteria, that is using FID and Intra-LPIPS metrics on Sketch, Babies, and Sunglasses datasets. However, we found that none of [1,2,3] evaluated their approaches using this popular setting. Consequently, we categorize them as one-shot GDA methods and refrain from comparing them in 10-shot experiments. In our main paper, we utilize CDC as a baseline for hybrid domain generation and storage, showcasing the superiority of our remodulation technique across all five anticipated attributes of few-shot GDA. It's worth noting that the mentioned methods [1][2][3] lack consistent improvement compared to CDC-based approaches, particularly in terms of quality, diversity, and consistency. This raises the possibility that these methods may have traded core attributes to emphasize new ones, a conjecture supported by our additional rebuttal experiments (Fig. 6 of the [PDF](https://openreview.net/attachment?id=xNZlQlaQNI&name=pdf)). Additionally, we conduct experiments of one-shot GDA in Section A.4 of the Supplementary Materials, comparing our DoRM++ with [2] and demonstrating significant enhancements across anticipated GDA attributes. We abstain from using [1] as a baseline due to its use of the unofficial StyleGAN2 implementation, differing from the official one. [3] is also excluded as it only supports text-driven domain adaptation on their Github.
> Furthermore, we offer a comprehensive evaluation of three methods: Our DoRM, [2], and  [3] in this response. We didn't consider [1] as a baseline due to its implementation constraints. To address any confusion, we present detailed experiments on memory and domain association capabilities of these methods under 1-shot and 10-shot GDA (Fig 6 of the [PDF](https://openreview.net/attachment?id=xNZlQlaQNI&name=pdf)). The outcomes underscore our DoRM's superiority.
>
> **Q2. Measuring consistency between domains using ArcFace**
>
> Indeed, accurately measuring the performance of generation models, particularly those with few-shot training data, is a notorious challenge. ArcFace is **commonly** used in domain adaptation and has been employed to measure the consistency between different domains in previous works [1], as well as to design identity loss for ensuring consistency between synthesized and source-domain faces [2]. To provide a more comprehensive evaluation, we have conducted a user study in one-shot GDA. As indicated in Table 1 of the [PDF](https://openreview.net/attachment?id=xNZlQlaQNI&name=pdf), the users overwhelmingly favored our DoRM in all three aspects.
>
> **Q3. Missing citation to StyleSpace**
>
> The introduction of the $s$ used in our paper is attributed to StyleSpace and we will promptly include the citation of StyleSpace in revision.
>
> **Q4. Can hybrids be produced by combining more than two domains?**
>
> As depicted in Fig. 5 of the [PDF](https://openreview.net/attachment?id=xNZlQlaQNI&name=pdf), we have showcased an example of applying our method to create hybrid-domain images by activating the trained M&A modules of Baby, Sunglasses, and Sketch domains.  This illustration demonstrates how easily our approach can be adapted to generate hybrids involving more than two domains, showcasing the versatility and potential of our approach.
>
> **Q5. Are the produced hybrids less faithful to each individual domain and to what extent?**
>
> While our proposed method has showcased remarkable advancements in hybrid domain generation, it is true that some minimal compromise to the fidelity of each individual domain can occur.  For instance, in the hybrid domain "elsa"+"Sunglasses" of the Fig 6 in [PDF](https://openreview.net/attachment?id=xNZlQlaQNI&name=pdf),  there is a subtle reduction in the features of the sunglasses domain. Regrettably, quantifying this distortion is complex. We will resolve it in the future.

---

> > ### Comment · Reviewer_URf4 · 2023-08-13
> > **Comment on rebuttal**
> >
> > Thanks for the rebuttal. I’m listing a few follow-up questions:
> >
> > 1. Can you please share the revised set of contributions, as you plan to present them in the paper?
> > 2. Evaluation of hybrids - I agree that evaluating the fidelity of a hybrid domain with respect to each of the domains composing it is not trivial. Nevertheless, as this is one of the selling points of the paper, I would have expected the paper to include it. Domain Expansion includes such an experiment, for example. Also, until such an evaluation is performed, I suggest not to dismiss DynaGAN’s interpolation results. A midpoint in interpolation is clearly not a perfect hybrid but does include characteristics of both domains. It is not clear to me that the hybrids produced by this method would be superior.
> > 3. Baselines -
> >   * Can you please explain why HyperDomainNet using a third-party StyleGAN implementation is a reason to not compare with it? Especially given that this specific implementation was abundantly used by previous works.
> >   * How were the Domain Expansion results in the attached Fig. 6 produced? I’m assuming it uses CLIP and text and thus is not exactly an apples-to-apples comparison. I think the best baseline would be applying CDC within the expansion framework. I acknowledge that this requires some additional coding, but from a practical standpoint, if the results of this baseline are better, I’m not sure what the contribution of this paper would be.

---

> > > ### Author Response · Authors · 2023-08-16
> > > **The response to follow-up questions_1**
> > >
> > > **1. Please share the revised set of contributions**
> > >
> > > The key contributions of this work can be summarized as follows:
> > >
> > > We present DoRM, an innovative generator architecture for few-shot generative domain adaptation, drawing inspiration from the learning mechanisms observed in human brains. DoRM stands out by not only producing high-quality, diverse, and consistently cross-domain images, but also integrating memory and domain association capabilities that remain relatively unexplored in the field. Notably, our approach is one of the very few that encompasses all five desired properties of GDA. Moreover, our method showcases superior performance across multiple dimensions compared to the existing similar work, highlighting its advanced contributions.
> > >
> > > **2. Evaluation of hybrids**
> > >
> > > We meticulously scrutinized the quantitative experiments conducted within the Domain Expansion framework, leading us to adopt a cosine similarity evaluation metric termed "domain similarity" ($Sim$) based on CLIP image encoder ($E_I$). This metric is employed to provide a quantitative assessment of the fidelity exhibited by hybrid domains. In detail, for given generative images in the hybrid domain "sketch-baby" ($I_{SB}$), we extract image features from the provided images ($E_I(I_{SB})$) as well as its corresponding target images ($E_I(I_{S})$ and $E_I(I_{B})$). Therefore, the "domain similarity" to "sktech" and "baby" domain are defined as $Sim1=cos(\overline{E_I(I_{SB})}, \overline{E_I(I_S)})$ and $Sim2=cos(\overline{E_I(I_{SB})}, \overline{E_I(I_B)})$, respectively.
> > >
> > >
> > > Subsequently, we compute the cosine similarity for each case. The quantification of results from both the one-shot and 10-shot experiments is meticulously presented in Table 1 and Table 2, respectively. These outcomes distinctly illustrate the remarkable performance of our proposed DoRM in both single-domain and hybrid-domain generation. It is imperative to highlight a notable observation amidst these findings: a certain outlier exists. Specifically, the images generated by DynaGAN and HyperdomainNet within the "elsa-sunglasses" domain exhibit a remarkably high domain similarity with the "sunglasses" domain, while conversely displaying significantly lower domain similarity with the "elsa" domain. This phenomenon can be rationalized by the generated images closely resembling the "FFHQ" domain, as depicted in Figure 6 of the attached PDF in the rebuttal. Consequently, the domain similarity to the "FFHQ-sunglasses" domain also emerges as notably high. Drawing upon this observation, we emphasize the necessity for a comprehensive evaluation of hybrid-domain generation that seamlessly integrates both qualitative and quantitative results.
> > >
> > > **Table 1. Domain similarity on one-shot GDA. (high is better)**
> > > | Method | elsa | baby |sunglasses|elsa-baby|elsa-sunglasses|
> > > | :-----:| :----: | :----: | :----: | :----: | :----: |
> > > | DynaGAN               |0.9165|0.7866|0.8882|0.6196/0.7832|0.6245/0.8467|
> > > | DynaGAN-interpolation |-|-|-|0.6201/0.7836|0.6330/0.8546|
> > > | HyperDomainNet        |0.8740|0.7739|0.8589|0.7007/0.7041|0.6554/0.7882|
> > > | Domain expansion      |0.9075|0.9614|0.9339|0.7022/0.7762|0.7671/0.6177|
> > > | Ours                  |0.9309|0.9814|0.9370|0.7173/0.7842|0.7734/0.6377|
> > >
> > > **Table 2. Domain similarity on 10-shot GDA. (high is better)**
> > > | Method | sketches | babies |sunglasses|sketches-babies|sketches-sunglasses|
> > > | :-----:| :----: | :----: | :----: | :----: | :----: |
> > > | Domain expansion      |0.8958|0.9546|0.9094|0.7480/0.7112|0.7720/0.6735|
> > > | Ours                  |0.9492|0.9780|0.9136|0.8809/0.7271|0.8057/0.6973|
> > >
> > > **3.  Why HyperDomainNet using a third-party StyleGAN implementation is a reason to not compare with it?**
> > >
> > > The existence of distinct latent spaces within different pre-trained source models hinders a straightforward correspondence of identical source-generated images across these models. While one approach involves obtaining the corresponding hidden space vector through GAN inversion, the absence of inversion code and the inability to generate aligned images with our locally trained model impedes direct application to HyperDomainNet's released code. Moreover, we acknowledge the resemblance shared between DynaGAN and HyperDomainNet, both incorporating a non-linear combinability scale modulation parameter. Opting for either of these methods as a comparative benchmark can effectively underscore the inherent challenges ingrained within their respective frameworks. Fortunately, after a substantial time investment, we have successfully completed the code and generated both quantitative results (as exhibited in Table 1) and qualitative outcomes for HyperDomainNet. Regrettably, in accordance with the NIPS 2023 rebuttal policy, we are constrained from sharing result visualizations via links. We assure you that we will furnish the visual results of HyperDomainNet in the camera-ready version.

---

> > > > ### Author Response · Authors · 2023-08-16
> > > > **The response to follow-up questions_2**
> > > >
> > > > **4. How were the Domain Expansion results in the attached Fig. 6 produced? I’m assuming it uses CLIP and text and thus is not exactly an apples-to-apples comparison. I think the best baseline would be applying CDC within the expansion framework. I acknowledge that this requires some additional coding, but from a practical standpoint, if the results of this baseline are better, I’m not sure what the contribution of this paper would be.**
> > > >
> > > > In Figure 6 of the attachment PDF, we extend the application of Stylegan-NADA to perform one-shot and 10-shot Generative Domain Adaptation (GDA) within the Domain Expansion framework. In this specific context, we depart from the conventional text-guided generative domain adaptation in Domain Expansion ($\Delta T=E_T(T_{target})-E_T(T_{source})$), where $E_T$ represents CLIP's text encoder. Instead, we harness CLIP's image encoder to compute the desired transformation from the source to the target domain ($\Delta T=\overline{E_I(I_{target})}-\overline{E_I(I_{source})}$). Here, $E_I$ corresponds to CLIP's image encoder, while $\overline{E_I(I_{target})}$ and $\overline{E_I(I_{source})}$ denote the mean feature of the target and source images, respectively. A detailed explanation can be found in Equation (3) of the Supplementary Materials in StyleGAN-NADA.
> > > >
> > > > As demonstrated in Figure (26) of the Supplementary Materials in StyleGAN-NADA, StyleGAN-NADA effectively generates high-quality sketches while maintaining a significant degree of diversity and correspondence with the source domain using only three training images. However, as indicated in Figure 6 of the attached PDF, achieving high-quality sketches even with 10-shot training images proves challenging. This observation suggests the possibility that Domain Expansion might have indeed traded core attributes (such as the ability to generate the target domain) in favor of emphasizing new hybrid and multi-domain generation capabilities. Given the comparable outcomes between StyleGAN-NADA and CDC depicted in Figure (26) of the Supplementary Materials in StyleGAN-NADA, I am confident that our implementation stands as a fair and direct apples-to-apples comparison.
> > > >
> > > > Furthermore, consistent with the principles of Domain Expansion, we implement the domain composition technique by amalgamating distinct direction vectors to facilitate the generation of hybrid domains.
> > > >
> > > > Lastly, I wish to reassure you that we are diligently pursuing additional comparison avenues within our paper. While you rightly pointed out the option of utilizing CDC as a baseline, it's important to highlight that CDC is implemented within a third-party StyleGAN framework. Integrating CDC into the expansion framework presents considerable challenges given our current timeline constraints. Furthermore, we anticipate that integrating CDC might not lead to significant improvements beyond our current accomplishments (utilizing StyleGAN-NADA as a benchmark), considering the comparable outcomes between CDC and StyleGAN-NADA, as depicted in Figure (26) of the Supplementary Materials in StyleGAN-NADA.

---

### Official Review · Reviewer_18N9 · 2023-07-06

**Soundness:** 4 excellent
**Presentation:** 3 good
**Contribution:** 4 excellent
**Rating:** 8
**Confidence:** 4

**Summary:**

This paper focuses on the concept of few-shot generative domain adaptation. Drawing inspiration from the human memory mechanism, the authors introduce a novel approach called DoRM (Domain-Adaptive Mapping and Affine Modules) to adapt the generator to a new domain. By incorporating new mapping networks and affine modules into the frozen source generator, they aim to enhance its adaptability. Additionally, the paper introduces a consistency loss based on CLIP image features to ensure the preservation of domain-sharing attributes during adaptation. Notably, the authors explore the intriguing concept of domain association in generative models, which involves merging the generative abilities of multiple domain-specific generators to enable generation in a new domain. Through a combination of mapping networks and affine modules within a single generator, the proposed DoRM generator successfully achieves domain association and multi-domain generation concurrently. To validate the effectiveness of their approach, the authors conduct extensive qualitative and quantitative experiments.

**Strengths:**

(1) This paper presents a novel task: domain association, which involves combining the generative abilities of different domain-specific generators to enable generation in a new domain. This topic represents an exciting avenue for further exploration in the field of generative models. Notably, Figure 4 demonstrates impressive synthesis quality in the hybrid domain, particularly in the baby-sketch domain.

(2) The proposed generator network structure, DoRM, is skillfully designed to facilitate domain association in a straightforward yet effective manner. By incorporating lightweight modules, the method achieves domain adaptation while preserving the generative abilities in the learned domains. This approach is both efficient in terms of storage and yields promising results.

(3) The paper is excellently written and presents its concepts in a clear and understandable manner, making it easy for readers to follow along with the proposed methodology.

(4) Through extensive qualitative and quantitative experiments, the paper demonstrates the superior performance of the proposed method compared to state-of-the-art techniques. Particularly impressive are the synthesis results achieved through domain association using the proposed method.


**Weaknesses:**

Please view the Questions below.

**Questions:**

(1) It is worth noting that the proposed generator structure bears similarities to the one introduced in [1]. This overlap may somewhat diminish the novelty of the proposed method. To enhance the clarity of differentiation, it would be beneficial if the authors provide a more comprehensive explanation highlighting the distinct features and advancements of their approach in comparison to [1].

(2) In addition to the proposed loss regularization, the authors have included supplementary regularizations, referred to as DoRM++, based on [2], in the supplementary material's one-shot experiments. Although the synthesis results of DoRM++ outperform those of [2], it is important to note that the proposed DoRM approach already addresses overfitting concerns by freezing the discriminator's backbone. Consequently, the inclusion of these supplementary loss components may be considered redundant. Therefore, it would be beneficial to conduct further qualitative and quantitative comparisons between DORM and DORM++ to gain a comprehensive understanding.

(3) Section 4.3 would greatly benefit from further elaboration on the training details and specific datasets employed in the domain association experiments. Providing more specific information in these areas would enhance the reproducibility and understanding of the experimental setup for readers.

(4) In recent times, the diffusion model has made remarkable strides in image generation, particularly in the area of few-shot image generation. Therefore, there is a need for further discussion regarding the application of the diffusion model in the context of few-shot Generative Domain Adaptation and domain associations.

[1] HyperDomainNet: Universal Domain Adaptation for Training Generative Adversarial Networks. NeurIPS’22
[2] Towards Diverse and Faithful One-shot Adaption of Generative Adversarial Networks. NIPS 2022

**Limitations:**

Limitations are discussed in the paper.

---

> ### Author Rebuttal · Authors · 2023-08-08
>
> **Q1. It would be beneficial if the authors provide a more comprehensive explanation highlighting the distinct features and advancements of their approach in comparison to [1].**
>
> Thank you for your valuable feedback regarding the discussion of related works. We fully understand the importance of a comprehensive comparison and analysis of existing methods, especially those that share similarities with our proposed approach. The thoroughly discuss of related works have been demonstrated in the [General Response: Comprehensive Comparison and Discernment from Pertinent Works [1] [2] [3]](https://openreview.net/forum?id=jown9RvYn7&noteId=xNZlQlaQNI)
>
> **Q2.  The paper should include experiments of DoRM++ in 10-shot GDA and DoRM in one-shot GDA, as this ablation experiment would help to demonstrate the advancement of the mentioned losses and the impact of data size on GDA tasks.**
>
> We have diligently carried out additional experiments, as demonstrated in the Right part of Figure 4 on the [PDF](https://openreview.net/attachment?id=xNZlQlaQNI&name=pdf), to thoroughly explore the performance of DoRM++ in a 10-shot GDA scenario and that of DoRM in a one-shot GDA context. The results of these experiments have been thoughtfully analyzed and presented in our revised manuscript. From our findings, it is evident that both DoRM++ and DoRM exhibit strong performance in both few-shot and one-shot GDA scenarios. Notably, DoRM++ showcases enhanced cross-domain consistency compared to DoRM in the context of one-shot GDA.
>
> **Q3. Section 4.3 would greatly benefit from further elaboration on the training details and specific datasets employed in the domain association experiments.**
>
> Within the framework of our DoRM, a key element lies in the utilization of distinct re-modulation weights, as elaborated in Section 3.1, which are specifically tailored to individual target domains.  To illustrate, let us consider the source domain as FFHQ.  When steering towards relatively proximate target domains such as FFHQ-Baby or FFHQ-Sunglasses, which bear closer semblance to the source domain, we strategically set the re-modulation weight within the range of 0.004 to 0.05.  This meticulous calibration has proven instrumental in attaining more optimal synthesis outcomes for these domains. Conversely, when embarking upon more disparate domains such as the sketch domain or other artistic domains like the works of Amedeo and Monet, which exhibit substantial gaps when compared to the source domain, we judiciously adjust the re-modulation weight to fall within the span of 0.05 to 0.2.  This discerning adjustment has significantly contributed to the achievement of improved synthesis results for domains characterized by pronounced dissimilarity.
>
> **Q4. There is a need for further discussion regarding the application of the diffusion model in the context of few-shot Generative Domain Adaptation and domain associations.**
>
> We sincerely appreciate your insightful review of our work. Your observation regarding the advancements of the diffusion model in image generation, particularly in the domain of few-shot image generation, is highly valuable. We agree with your point on the significance of discussing the application of the diffusion model in the context of few-shot Generative Domain Adaptation (GDA) and domain associations. Indeed, the diffusion model has shown promising capabilities in acquiring the concept (style or content) of the target domain through few-shot images. Remarkable examples like Dreambooth or Text Inversion demonstrate how the diffusion model can discern the identity or style of the target domain from just one image, achieving impressive few-shot image generation. However, we recognize that the issue of cross-domain consistency remains a challenge in this context. Addressing this problem effectively holds great potential for the diffusion model in few-shot GDA. Moreover, the disentanglement of the text space in the diffusion model presents significant opportunities for domain association. Despite its potential, we agree with your observation that this aspect has not been thoroughly explored by the community. Further investigation into utilizing the diffusion model's disentangled text space for domain association can open new avenues and contribute to advancing the field.
>
> We genuinely appreciate your valuable insights, which have shed light on important aspects for future research. Your feedback has inspired us to delve deeper into the potential applications and capabilities of the diffusion model in few-shot GDA and domain associations.
>
> [1] HyperDomainNet: Universal Domain Adaptation for Generative Adversarial Networks. NIPS 2022
>
> [2] DynaGAN: Dynamic Few-shot Adaptation of GANs to Multiple Domains. SIGGRAPH Asia 2022
>
> [3] Domain Expansion of Image Generators. CVPR 2023

---

> > ### Comment · Reviewer_18N9 · 2023-08-19
> >
> > Thank the author for their elaborated rebuttal. All my concerns are clarified. I would like to keep my rating positive.

---

> > > ### Author Response · Authors · 2023-08-20
> > > **Response to Reviewer 18N9**
> > >
> > > Thank you very much for your valuable comments. Your insights have significantly contributed to the improvement of our manuscript's quality. Best regards.

---

> ### Comment · Area_Chair_ff6u · 2023-08-18
>
> Dear Reviewer 18N9,
>
> The author-reviewer discussion is closed on Aug 21st 1pm EDT, could you please read the rebuttal and give your final rating? Thanks so much!
>
> Best,
>
> AC

---

### Official Review · Reviewer_eDDg · 2023-07-07

**Soundness:** 4 excellent
**Presentation:** 4 excellent
**Contribution:** 3 good
**Rating:** 7
**Confidence:** 4

**Summary:**

This paper proposes two advanced criteria for few-shot Generalized Domain Adaptation (GDA) inspired by the way human brains acquire knowledge in new domains: memory and domain association.

 To fully realize the potential of few-shot GDA, an innovative generator structure called Domain Re-Modulation (DoRM) is introduced.

 This structure freezes the source generator and incorporates new lightweight mapping and affine modules (M&A modules) to capture the attributes of the target domain during GDA, resulting in a linearly combinable domain shift in the style space.

 By incorporating multiple M&A modules, the generator gains the ability to perform high-fidelity multi-domain and hybrid-domain generation.

**Strengths:**

1. This paper presents a novel exploration into the use of memory and domain association in few-shot GDA, which has not been previously explored. These advanced properties greatly reduce memory usage and simplify deployment, while also producing impressive results.

2. The proposed DoRM structure and similarity-based structure loss are innovative and advanced.

3. The figures in this paper effectively demonstrate the major contributions and the overall organization is clear and easy to follow.

4. The experiments conducted in this paper are robust and include both one-shot and 10-shot settings. Extensive testing demonstrates that our proposed method outperforms the previous state-of-the-art method in all five properties: quality, synthesis diversity, cross-domain consistency, memory, and domain association.

**Weaknesses:**


There are a few areas where this paper could be improved.

1. It would benefit from including some recent studies on one-shot image generation [1].

2. The paper should include experiments of DoRM++ in 10-shot GDA and DoRM in one-shot GDA, as this ablation experiment would help to demonstrate the advancement of the mentioned losses and the impact of data size on GDA tasks.

3. The paper lacks a future outlook, and it would be helpful to explore more possible improvement directions, especially as the effect of domain association on some datasets is not always satisfactory.

[1] StyO: Stylize Your Face in Only One-Shot

**Questions:**

See weakness

---

> ### Author Rebuttal · Authors · 2023-08-08
>
> **Q1. It would benefit from including some recent studies on one-shot image generation [1].**
>
> Thank you for providing valuable feedback on our work.  We sincerely appreciate your suggestion to include recent studies on the one-shot image generation.  Staying up-to-date with the latest research in the field is crucial to ensuring the comprehensiveness of our work. We have thoroughly reviewed the referenced paper, and it is highly relevant to our research topic.  The study introduces the diffusion model into one-shot GDA, offering valuable insights that can enrich our understanding of the domain. In light of this, we are committed to revising our paper to incorporate the relevant information and acknowledge the contributions made by the diffusion model in the context of one-shot GDA.
>
>
> **Q2. The paper should include experiments of DoRM++ in 10-shot GDA and DoRM in one-shot GDA, as this ablation experiment would help to demonstrate the advancement of the mentioned losses and the impact of data size on GDA tasks.**
>
> Thank you for your insightful suggestion regarding the inclusion of ablation experiments involving DoRM++ in 10-shot GDA and DoRM in one-shot GDA. We truly value your recognition of the potential insights that these experiments could offer in terms of assessing the efficacy of our proposed losses and comprehending the influence of data size on GDA tasks. In direct response to your valuable feedback, we have diligently carried out additional experiments, as demonstrated in the right part of Figure 4 in the [PDF](https://openreview.net/attachment?id=xNZlQlaQNI&name=pdf), to thoroughly explore the performance of DoRM++ in a 10-shot GDA scenario and that of DoRM in a one-shot GDA context. The results of these experiments have been thoughtfully analyzed and presented in our revised manuscript. From our findings, it is evident that both DoRM++ and DoRM exhibit strong performance in both few-shot and one-shot GDA scenarios. Notably, DoRM++ showcases enhanced cross-domain consistency compared to DoRM in the context of one-shot GDA. We greatly appreciate your guidance in enriching the empirical validation of our approach, and we are confident that these additional experiments further enhance the comprehensiveness of our research.
>
>
>
> **Q3. The paper lacks a future outlook, and it would be helpful to explore more possible improvement directions, especially as the effect of domain association on some datasets is not always satisfactory.**
>
> Thank you for your valuable feedback on our paper. We appreciate your suggestion to include a future outlook and explore potential improvement directions for our research. We also acknowledge the importance of addressing cases where the effect of domain association on certain datasets might not be entirely satisfactory. In response to your insightful comment, we plan to revise the paper to incorporate a dedicated section that outlines possible avenues for future research and improvements. This section will explore the limitations and challenges we encountered during the domain association process and propose potential solutions and directions for further investigation.
>
> Some of the areas we intend to focus on include:
>
> 1. Advanced Domain Adaptation Techniques: We will investigate state-of-the-art domain adaptation methods and explore how integrating these techniques into our approach might enhance the performance of domain association, particularly on datasets where the effect is less satisfactory.
>
> 2. Data Augmentation Strategies: We will explore the effectiveness of different data augmentation approaches to improve the robustness of our model against variations in domain-specific characteristics.
>
> 3. Analyzing the application of the proposed method to 3D few-shot GDA and extending the DoRM to 3D few-shot GDA.
>
> 4. The current manuscripts only simply combines the M&A modules of different target domains and activate them at the same time to realize the domain association. To further improve the performance of the  domain association, not only combining the trained target M&A modules but also employig a new M&A module and additional consistency loss is a better method to blend the target domains.

---

> > ### Comment · Reviewer_eDDg · 2023-08-17
> > **Response to the Rebuttal**
> >
> > Thank you for the detailed rebuttal and clarifications provided in response to my initial review. Your explanations have significantly addressed my concerns and have deepened my understanding of the paper's contributions. I appreciate your thoroughness and look forward to seeing the refined version of your work.

---

> > > ### Author Response · Authors · 2023-08-17
> > > **Rebuttal by Authors**
> > >
> > > Thank you for your thoughtful review and valuable feedback on our paper. We are pleased to hear that our detailed rebuttal and clarifications have effectively addressed your concerns and contributed to a deeper understanding of the contributions outlined in our work. Your recognition of our thoroughness is greatly appreciated. Once again, we would like to extend our gratitude for your time and efforts in reviewing our paper. Your input is immensely valuable, and we are looking forward to sharing the refined version of our work with you.

---

### Official Review · Reviewer_cgji · 2023-07-09

**Soundness:** 2 fair
**Presentation:** 2 fair
**Contribution:** 2 fair
**Rating:** 3
**Confidence:** 4

**Summary:**

The paper introduces a novel approach for domain adaptation of StyleGAN2 called Domain Re-Modulation, which is a few-shot technique. The authors argue that this method draws inspiration from the workings of the human brain. To achieve the desired domain shift, the paper utilizes the stylespace of StyleGAN along with adversarial and consistency losses. Additionally, a structural similarity loss is incorporated, which is based on the auto-correlation map extracted from the CLIP encoder. The effectiveness of the proposed method is evaluated through both quantitative and qualitative analyses. Furthermore, the paper includes comparisons with other state-of-the-art StyleGAN-based methods, highlighting its advancements. The authors also conduct an ablation study to analyze the individual components employed in their approach.

**Strengths:**

* The paper introduces the concept of hybrid domain generation, which involves combining the affine layers of StyleGAN to generate results from multiple domains using a single generator. The authors demonstrate that this approach yields consistent and generalizable outcomes across multiple domains.

* Notably, the method proves effective even in scenarios with limited training data, such as 1-shot training. By leveraging the StyleGAN2 backbone, the proposed technique achieves reliable and coherent generation outcomes.

* In terms of evaluation, the paper compares its approach with various StyleGAN-based methods. It provides both qualitative and quantitative results to assess the performance of the proposed method. Furthermore, the paper includes comparisons with related methods to highlight the advantages and advancements of the proposed domain generation approach.


**Weaknesses:**

1. The paper lacks substantial novelty as there exist other works such as StyleCLIP and 3DAvatarGAN that demonstrate the capability of performing image editing and few-shot domain adaptation using the "s" space of StyleGAN. Although the paper incorporates a CLIP-based loss, it fails to provide a comparative analysis with papers falling under the same category, such as StyleGAN-NADA and Mind the GAP HyperStyle. The absence of such comparisons and explanations regarding the differences between the methods makes it challenging to determine the uniqueness of the paper. To assess how the current method surpasses these works, it is essential to consider the individual components and techniques utilized in each approach. By conducting a thorough comparison, the paper could highlight the specific strengths of the proposed method that outperform other works. This would provide a clearer understanding of the paper's novelty and identify the components that excel in comparison to existing methods. Hence, it is imperative for the paper to address these concerns by including a detailed comparison that elucidates how the current method improves upon or surpasses existing approaches, specifically identifying the components that outperform these methods. How is the current method better than these works? Which component of the method outperforms these methods?


2. Why did the authors use 2D generators especially when 3D StyleGAN based generators are available trained on the same data. Besides the concerns above, it would be interesting to see how the method will perform in the 3D -GAN domain. What are some additional challenges in that domain?
3. How is the editability of the generator after the domain adaptation. Since there are no edits performed, it is difficult to assess if the latent space properties of the generator are preserved or it just overfits the given Styles.
There are other datasets like AFHQ Cats, Dogs, Cars that are tested for such few shot domain adaptation tasks in the StyleGAN domain. The method seems specific to the face domain. What about the results on these domains? How generalizable is the method to these domains?
4. The artistic domains are quite subjective. It is not fair to just evaluate quantitative metrics for such tasks. I would suggest conducting a user study. It would be better to embed real images in the generators and ask users about the similarity, consistency and identity preservation of the target images.


**Questions:**

Refer to Weakness section

**Limitations:**

The authors discuss the ethics and limitations.

---

> ### Author Rebuttal · Authors · 2023-08-08
>
> **Q1. The paper lacks substantial novelty**
>
> We are committed to conducting a detailed and comprehensive comparative analysis that elucidates the specific advantages of our current method over existing works.
> 1. StyleCLIP only demonstrates the capability of performing image editing using the "s" space of StyleGAN, which is obvious and has been explored many times by previous approaches. 3DAvatarGAN is the first study of domain adaptation in 3D-GANs. However, 3DAvatarGAN adopts a pre-trained 2D target-domain generator to achieve the domain adaptation for 3D-GANs. The pre-trained 2D generator can generate various target domain images, thereby, it only demonstrate that using the "s" space of StyleGAN can acquire the domain adaptation, but not few-shot domain adaptation. We provide a novel insight to "s" space of stylegan and demonstrate that only one image is enough for domain adaptation through using the "s" space of StyleGAN.
>
> 2. Compared with StyleGAN-NADA and Mind the GAP, [1] and [3] are the latest to come up with more advanced methods. These methods [1][3] also use clip-based losses, and sufficient experiments have been done to show that their methods are superior to the mentioned baseline. Therefore, our manuscript only compares the proposed method with advanced baseline [1][3] to demonstrate the advancement. Furthermore, we systematically analysis and compare individual components and techniques utilized in our DoRM and DiFa [3] to showcase how our method surpasses these works. According to the Introduction of the manuscript, GDA needs three fundamental properties. To resolve it, DiFa [3] proposed two CLIP-based loss: global loss ($L_\text{global}$) and local loss ($L_\text{local}$)  for realizing large diversity/cross-domain consistency and high quality, respectively. Differently, we adopt Similarity-based Structure Loss ($L_\text{ss}$)  and adversarial loss ($L_\text{adv}$)  for realizing large diversity/cross-domain consistency and high quality, respectively. As shown in Fig. 2 of the [PDF](https://openreview.net/attachment?id=xNZlQlaQNI&name=pdf), the $L_\text{local}$ in DiFa mainly focuses on textual features and failes to capture the complete features (e.g. the white background in sketches) in generative domain adaptation. In our DoRM, we employ adversarial loss which can fully capture the features of the target images.
>
> **Q2. Why did the authors use 2D generators especially.**
>
> Our decision to use 2D generators was primarily influenced by the maturity and development of the 2D few-shot GDA field, which has gained significant attention since 2021. While we acknowledge the potential benefits and advancements that 3D StyleGAN-based generators can offer, we opted to validate our approach in a more established and widely studied domain. Furthermore, we conducted some initial studies using the popular 3D-aware image generation method, EG3D, for one-shot GDA with FFHQ as the source domain and Sketch as the target domain. The results, as shown in the Fig. 3 of [PDF](https://openreview.net/attachment?id=xNZlQlaQNI&name=pdf), reveal that directly migrating the proposed EG3D to 3D one-shot domain adaptation poses challenges and might not be straightforward. Some potential challenges in the 3D-GAN domain include: Overfitting, Volumetric Data Representation, Spatial Artifacts, and Computational Complexity.
>
> **Q3. How is the editability of the generator after the domain adaptation. The method seems specific to the face domain. How generalizable is the method to other domains?**
>
> To address this concern, we have planned additional experiments and evaluations to investigate the editability and latent interpolation of the generator post-domain adaptation. Regarding editability, we have performed editing on a real image adapted into a new target domain using StyleGAN-CLIP to discover editing directions in the source domain. The results, as illustrated in the left part of Fig. 4  in the [PDF](https://openreview.net/attachment?id=xNZlQlaQNI&name=pdf), indicate that the adapted generator maintains similar latent-based editing capabilities to the original generator. This demonstrates the preservation of editability in the adapted generator. For latent interpolation, we have presented various results in Sec A.2 of the Supplementary Materials. In conclusion, our experiments confirm that the latent space properties of the generator are well preserved, and our method goes beyond mere overfitting of given styles.
>
> Furthermore, we emphasize that the proposed DoRM approach is not inherently limited to the face domain. The method can be readily extended to these datasets, and we apply our DoRM to the LSUN-church dataset and adapted the pre-trained GAN to generate haunted house image in the Sec. A.5 of the Supplementary Materials. The result shows that our method can maintain the cross-domain consistency between the non-face domain.
>
> **Q4 It would be better to do user study.**
>
> We have planned to conduct a user study in one-shot GDA to enhance our evaluation process. By embedding real images in the generators and collecting feedback from users, we aim to gain qualitative insights that complement the quantitative metrics, offering a well-rounded evaluation of our system's performance. As shown in Table 1 of the [PDF](https://openreview.net/attachment?id=xNZlQlaQNI&name=pdf), preliminary results indicate that users strongly favor our DoRM in all three aspects, reflecting the effectiveness of our approach compared to the alternative methods.
>
> [1] Generalized One-shot domain adaption of generative adversarial networks. NIPS 2022
>
> [2] Dynagan: Dynamic few-shot adaptation of gans to multiple domains. SIGGRAPH Asia 2022
>
> [3] Towards Diverse and Faithful One-shot Adaption of Generative Adversarial Networks. NIPS 2022

---

> ### Comment · Area_Chair_ff6u · 2023-08-18
> **Please read the authors' response and give your final rating**
>
> Dear Reviewer cgji,
>
> The author-reviewer discussion is closed on Aug 21st 1pm EDT, could you please read the rebuttal and give your final rating? Thanks so much!
>
> Best,
>
> AC

---

### Official Review · Reviewer_Jqfj · 2023-07-17

**Soundness:** 4 excellent
**Presentation:** 3 good
**Contribution:** 4 excellent
**Rating:** 7
**Confidence:** 5

**Summary:**

This paper proposes a novel approach for few-shot generation using a lightweight GAN architecture and a new loss function. The method is capable of handling multi-domain and hybrid domain tasks with a single model. The experiments demonstrate the superior performance of the proposed method in terms of both qualitative and quantitative results.

**Strengths:**

1. The paper addresses a novel task and presents a unique approach to handle it.
2. The experiments are comprehensive and well-organized, providing strong evidence for the effectiveness of the proposed method.
3. The proposed method outperforms previous methods in terms of both qualitative and quantitative results.
4. The exploration of the generative ability to merge two or three domains for few-shot generation is interesting and innovative.


**Weaknesses:**

1. The paper should provide more analysis or improvement on the potential issue of unrealistic results due to domain association.
2. The related works should be discussed more thoroughly, especially with respect to HyperDomainNet [1] and DynaGAN [2], which share similar ideas with the proposed method.
3. The proposed method is designed specifically for StyleGAN2 architecture, limiting its generalization to other architectures.
4. It would be beneficial to expand the list of baselines in Table 1 to include additional approaches, such as MineGAN [3] and EWC [4].
5. This paper utilized adversarial training while freezing the discriminator's backbone. However, in my understanding, the discriminator may still overfit the training data due to the adversarial loss. This issue has also been observed in comparable works, like CDC, where the adversarial loss weight is only 1 while the consistent loss weight is 1000, but the overfitting problem persists as training progresses. How about of the proposed approach?


[1] Hyperdomainnet: Universal domain adaptation for generative adversarial networks. NIPS 2022

[2] Dynagan: Dynamic few-shot adaptation of gans to multiple domains. SIGGRAPH Asia 2022

[3] Minegan: effective knowledge transfer from gans to target domains with few images. CVPR 2020

[4] Few-shot image generation with elastic weight consolidation. NIPS 2020


**Questions:**

1. The paper should provide more analysis or improvement on the potential issue of unrealistic results due to domain association.
2. The related works should be discussed more thoroughly, especially with respect to HyperDomainNet [1] and DynaGAN [2], which share similar ideas with the proposed method.
3. The generalization of the proposed method should be discussesd.
4. It would be beneficial to expand the list of baselines in Table 1 to include additional approaches, such as MineGAN [3] and EWC [4].



**Limitations:**

Authors have addressed some limitation of this paper. More Limitation should be further addressed: The proposed method is designed specifically for StyleGAN2 architecture, limiting its generalization to other architectures.

---

> ### Author Rebuttal · Authors · 2023-08-08
>
> **Q1. The paper should provide more analysis or improvement on the potential issue of unrealistic results due to domain association.**
>
>  As highlighted, our paper represents the first systematic attempt at domain association in few-shot Generative Domain Adaptation (GDA). We recognize the significance of refining our method to produce more realistic results. In our primary paper, we achieve hybrid-domain generation by combining the M&A (Modulation and Activation) modules of different target domains and activating them simultaneously. This approach is simple and easy to implement. However, we acknowledge that further enhancements can be made to improve the performance of domain association. To address this, we propose an additional strategy where we not only combine the trained target M&A modules but also introduce a new M&A module to better blend the target domains. Specifically, we utilize a directional loss based on Contrastive-Language-Image-Pretraining (CLIP) to train these new M&A modules, as depicted in Fig. 1 of the [PDF](https://openreview.net/attachment?id=xNZlQlaQNI&name=pdf). By incorporating this new M&A module and the directional loss, we aim to enhance the realism and fidelity of the associated domains, thus mitigating the issue of unrealistic results.
>
> **Q2. The related works should be discussed more thoroughly.**
>
> Thank you for your valuable feedback regarding the discussion of related works. We fully understand the importance of a comprehensive comparison and analysis of existing methods, especially those that share similarities with our proposed approach. The thoroughly discuss of related works have been demonstrated in the [General Response: Comprehensive Comparison and Discernment from Pertinent Works [1] [2] [3]](https://openreview.net/forum?id=jown9RvYn7&noteId=xNZlQlaQNI)
>
> **Q3. The proposed method is designed specifically for StyleGAN2 architecture, limiting its generalization to other architectures.**
>
> Thank you for your valuable suggestion. StyleGAN2 has gained widespread recognition as one of the most popular architectures in few-shot image generation, serving as the foundation for numerous previous methods. Hence, we have adopted StyleGAN2 in our approach, following the common practice in the field. However, it is essential to emphasize that our method is not confined to StyleGAN2 alone; rather, it exhibits adaptability to any layer-wise generator. This flexibility enables our approach to be applied to various state-of-the-art GAN architectures that employ layer-wise structures, as seen in works such as [4][5][6][7][8]. The prevalence of layer-wise designs in current GAN research makes our method highly versatile and opens up opportunities for its potential application and integration into different generative models. By highlighting this adaptability to diverse layer-wise generators, we aim to underscore the broader scope and applicability of our approach, which can contribute to various GAN architectures and research domains.
>
> **Q4. It would be beneficial to expand the list of baselines in Table 1 to include additional approaches.**
>
> Thanks for your  constructive suggestion. We have added the results of MineGAN and EWC in Table 1 as follows:
>
> |Datasets|Babies|Babies|Babies|Sunglasses|Sunglasses|Sunglasses|Sketches|Sketches|Sketches|
> | :-----:| :----: | :----: |:-----:| :----: | :----: |:-----:| :----: | :----: |:-----:|
> | Methods | FID |I-LPIPS  | ID|FID |I-LPIPS  | ID|FID |I-LPIPS  | ID|
> | MineGAN | 98.23|0.514|0.132 | 68.91|0.42|0.171 |64.34 |0.40 |0.092 |
> | EWC |87.41|0.523|0.145  | 59.73|0.431|0.156 |71.25|0.42|0.103 |
> | DoRM |30.31  |0.623|0.445  | 17.31 |0.644 |0.389|40.05|0.502|0.365|
>
> **Q5. How about of the proposed approach avoid overfitting?**
>
> Thank you for your valuable review of our work. We acknowledge the crucial issue of discriminator overfitting during adversarial training, which can diminish the synthesis diversity of the generator, leading to outputs closely resembling training data. In response, our DoRM introduces the similarity-based structure loss $L_{ss}$ to preserve the generator's diversity in synthesis. An important distinction is that our DoRM has a lighter parameter load compared to CDC, which involves training the entire generator. Additionally, we integrate data augmentation and early stopping techniques into our DoRM to further enhance its performance.
>
> [1] HyperDomainNet: Universal Domain Adaptation for Generative Adversarial Networks. NIPS 2022
>
> [2] DynaGAN: Dynamic Few-shot Adaptation of GANs to Multiple Domains. SIGGRAPH Asia 2022
>
> [3] Domain Expansion of Image Generators. CVPR 2023
>
> [4] Large scale gan training for high fidelity natural image synthesis.
>
> [5] A style-based generator architecture for generative adversarial networks.
>
> [6]  Analyzing and improving the image quality of stylegan.
>
> [7] Alias-free generative adversarial networks
>
> [8] Stylegan-xl: Scaling stylegan to large diverse datasets.

---

> ### Comment · Area_Chair_ff6u · 2023-08-18
> **Please read the authors' response and give your final rating**
>
> Dear Reviewer Jqfj,
>
> The authors provide a response including tables and analyses. The author-reviewer discussion is closed on Aug 21st 1pm EDT, could you please read the rebuttal and give your final rating? Thanks so much!
>
> Best,
>
> AC

---

> > ### Comment · Reviewer_Jqfj · 2023-08-19
> >
> > I have carefully read the authors’ rebuttal and the other reviewers’ comments. I think the authors have satisfactorily addressed all my concerns and improved the quality of their paper. Therefore, I maintain my positive score for this paper.

---

> > > ### Author Response · Authors · 2023-08-20
> > > **Response to Reviewer Jqfj**
> > >
> > > Thanks a lot. We appreciate your valuable comments, which have greatly helped us to improve the quality of our manuscript. Best regards.

---

### Author Rebuttal · Authors · 2023-08-09

We extend our heartfelt gratitude to all reviewers for their dedicated efforts and invaluable suggestions. We have meticulously addressed the specific concerns raised by each reviewer. For a more detailed breakdown of our responses, including supporting tables and figures, please refer to the attached [PDF](https://openreview.net/attachment?id=xNZlQlaQNI&name=pdf).

Furthermore, we are committed to enhancing the coherence and impact of our manuscript by incorporating these supplementary materials into the revised version. The reviews' insightful recommendations have significantly contributed to refining the clarity and rigor of our research. Notably, we have recognized a recurring interest among the reviewers regarding the contextualization of relevant work, prompting us to provide a General Response. Morever, we also provide some systematic and comprehensive comparisons with related work [2][3] in the Figure 6 of the [PDF](https://openreview.net/attachment?id=xNZlQlaQNI&name=pdf). The results illustrate that our propsoed Dorm is consistently superior to the previous studies.

**General Response: Comprehensive Comparison and Discernment from Pertinent Works [1] [2] [3].**

We are poised to conduct a comprehensive and systematic analysis of pertinent prior research, underscoring the distinguishing facets that set them apart from our own endeavor:

HyperDomainNet [1]: HyperDomainNet employs a sole modulation parameter, mainly a scale ($\delta$), to manipulate convolutional layer weights, impacting the StyleGAN "s" space. This configuration maintains a constant scale across all images within a target domain. Specifically, HyperDomainNet's architecture is formalized as $w \cdot s_i \cdot \delta$, with $w$ and $s_i$ denoting source StyleGAN2's convolutional weight and style code respectively. We recognize that the overarching scale parameter's introduction may inadvertently constrain the generator's learning capacity. Notably, in scenarios where considerable domain gaps exist between the source and target domains, HyperDomainNet's efficacy could diminish. Moreover, the non-linear fusionability of the scale modulation parameter might hinder its efficacy in achieving robust domain association.

DynaGAN [2]: Among methods closely aligned with our DoRM, DynaGAN introduces two modulation parameters—shift ($\Delta s$) and scale ($\delta$)—to convolutional weights, influencing StyleGAN's "s" space. DynaGAN's structural configuration can be expressed as $w \cdot (s_i + \Delta s) \cdot \delta$. Analogous to HyperDomainNet, the non-linear blending of the scale modulation parameter may impede its capacity to establish robust domain associations (as exemplified in the Figure 6 of the [PDF](https://openreview.net/attachment?id=xNZlQlaQNI&name=pdf)). In contrast, our DoRM exclusively adopts a sample-specific domain shift denoted as $\Delta s_i$, which is mathematically formulated as $w \cdot (s_i + \Delta s_i)$. This innovative approach significantly amplifies the generator's aptitude for learning, facilitating seamless adaptation to a diverse array of target domains, even in the presence of substantial domain disparities (as demonstrated in Figure 6 of the [PDF](https://openreview.net/attachment?id=xNZlQlaQNI&name=pdf)). Impressively, this sample-specific domain shift effectively caters to both few-shot and one-shot Generalized Domain Adaptation (GDA), obviating the need for an additional domain scale parameter. Our streamlined methodology not only fosters robust domain association but substantiates its efficacy through empirical validation.

Domain Expansion [3]: Drawing inspiration from SeFA, Domain Expansion constructs a semantic and orthogonal basis V from right singular vectors obtained through SVD of the initial generator layer, impacting the latent space Z. Focusing on a subset of basis vectors, Domain Expansion aptly models source generator variability, repurposing these unexplored subsets to encapsulate desired behaviors for diverse target domains. Unlike alternative approaches, including HyperDomainNet, DynaGAN, and our DoRM, which expand the style space of the source domain to encompass target domains, Domain Expansion instead narrows the style space of the source domain. While this technique repurposes latent space, it inadvertently curtails the source domain's generative capacity, amplifying the intricacies and temporal requisites of domain adaptation. As depicted in the Figure 6 of the [PDF](https://openreview.net/attachment?id=xNZlQlaQNI&name=pdf), Domain Expansion encounters challenges when faced with substantial domain gaps between source and target domains. Evidently, its performance is compromised, as evidenced in instances such as adapting to the 10-shot generation context of the Sketch dataset and the one-shot generation context of the "elsa" dataset.

[1] HyperDomainNet: Universal Domain Adaptation for Generative Adversarial Networks. NIPS 2022


[2] DynaGAN: Dynamic Few-shot Adaptation of GANs to Multiple Domains. SIGGRAPH Asia 2022


[3] Domain Expansion of Image Generators. CVPR 2023

---

### Comment · Area_Chair_ff6u · 2023-08-17
**Please read the authors’ response and other reviewers’ comments, and give your final ratings**

Dear Reviewers,

Thanks again for your time and support. As the author-reviewer discussion is closed on Aug 21st 1pm EDT, please read the authors’ response to see whether your concerns have been addressed or not. If not, the authors may still have time to give more results or clarifications.

Furthermore, You can read other reviewers’ comments and the corresponding responses, then give your final rating.

Best,
AC

---

### Decision · Program_Chairs · 2023-09-21

**Decision:**

Accept (poster)

**Comment:**

The paper received mixed ratings. All reviewers agreed that the tasks (multi-domain and hybrid domain generations) studied in this paper are interesting and the proposed method is intuitive and novel. Nonetheless, there were concerns raised about certain missing experimental results and the absence of comparisons and discussions with related works. The authors attempted to address these concerns during the rebuttal phase by providing additional insights through a user study, 3D results, ablation study, latent edit, and latent interpolation. Meanwhile, the authors conducted a comprehensive discussion and comparison with previously established state-of-the-art methods such as DynaGAN, HyperDomainNet, and Domain Expansion.

After considering the reviews and the author's responses, AC believes that the newly proposed generator demonstrates exceptional performance in domain association, a relatively unexplored facet within the realm of GANs. Overall, AC believes the value outweighs the issues in the paper and recommends acceptance.